# High-throughput, single-particle tracking reveals nested membrane domains that dictate KRas[G12D] diffusion and trafficking

Yerim Lee[1,2], Carey Phelps[1,2], Tao Huang[1,2], Barmak Mostofian[1,2], Lei Wu[1,2,3], Ying Zhang[1,2], Kai Tao[1,2], Young Hwan Chang[1,2], Philip JS Stork[4], Joe W Gray[1,2], Daniel M Zuckerman[1,2]*, Xiaolin Nan[1,2,5]*

[1]Department of Biomedical Engineering, Oregon Health and Science University, Portland, United States; [2]OHSU Center for Spatial Systems Biomedicine (OCSSB), Oregon Health and Science University, Portland, United States; [3]Department of Oral Maxillofacial-Head Neck Oncology, School and Hospital of Stomatology, Wuhan University, Wuhan, China; [4]Vollum Institute, Oregon Health and Science University, Portland, United States; [5]Knight Cancer Early Detection Advanced Research (CEDAR) Center, Oregon Health and Science University, Portland, United States

**Abstract** Membrane nanodomains have been implicated in Ras signaling, but what these domains are and how they interact with Ras remain obscure. Here, using single particle tracking with photoactivated localization microscopy (spt-PALM) and detailed trajectory analysis, we show that distinct membrane domains dictate KRas[G12D] (an active KRas mutant) diffusion and trafficking in U2OS cells. KRas[G12D] exhibits an immobile state in ~70 nm domains, each embedded in a larger domain (~200 nm) that confers intermediate mobility, while the rest of the membrane supports fast diffusion. Moreover, KRas[G12D] is continuously removed from the membrane via the immobile state and replenished to the fast state, reminiscent of Ras internalization and recycling. Importantly, both the diffusion and trafficking properties of KRas[G12D] remain invariant over a broad range of protein expression levels. Our results reveal how membrane organization dictates membrane diffusion and trafficking of Ras and offer new insight into the spatial regulation of Ras signaling.

*For correspondence:
zuckermd@ohsu.edu (DMZ);
nan@ohsu.edu (XN)

**Competing interests:** The authors declare that no competing interests exist.

## Introduction

The plasma membrane has a complex and dynamic landscape that helps shape how diverse membrane-localized signaling molecules behave (*Grecco et al., 2011*; *Staubach and Hanisch, 2011*; *Simons and Toomre, 2000*; *Schmick and Bastiaens, 2014*; *Varma and Mayor, 1998*; *Garcia-Parajo et al., 2014*). Among others, the Ras small GTPases are prototypical examples of signaling molecules whose biological activities are directly regulated by the membrane (*Zhou and Hancock, 2015*; *Abankwa et al., 2010*). While biochemical aspects of how Ras interacts with downstream effectors such as Raf have been well studied (*Simanshu et al., 2017*; *Cox and Der, 2010*), the mechanisms through which the biological membrane defines the signaling activity and specificity of Ras are still poorly understood. Recent studies by us and others suggest that Ras signaling may involve the formation of multimers (dimers and/or clusters) in a membrane-dependent manner (*Güldenhaupt et al., 2012*; *Nan et al., 2015*; *Spencer-Smith et al., 2017*; *Ambrogio et al., 2018*; *Prior et al., 2003*; *Inouye et al., 2000*; *Plowman et al., 2005*), and that partitioning of Ras into nanoscopic membrane domains and interactions with scaffold proteins or structures likely constitute critical steps to Ras multimer formation and signaling (*Zhou et al., 2015*; *Chung et al., 2018*; *Belanis et al., 2008*; *Shalom-Feuerstein et al., 2008*; *Chen et al., 2016*). While previous high-

**eLife digest** The Ras family of proteins play an important role in relaying signals from the outside to the inside of the cell. Ras proteins are attached by a fatty tail to the inner surface of the cell membrane. When activated they transmit a burst of signal that controls critical behaviors like growth, survival and movement. It has been suggested that to prevent these signals from being accidently activated, Ras molecules must group together at specialized sites within the membrane before passing on their message. However, visualizing how Ras molecules cluster together at these domains has thus far been challenging. As a result, little is known about where these sites are located and how Ras molecules come to a stop at these domains.

Now, Lee et al. have combined two microscopy techniques called 'single-particle tracking' and 'photoactivated localization microscopy' to track how individual molecules of activated Ras move in human cells grown in the lab. This revealed that Ras molecules quickly diffuse along the inside of the membrane until they arrive at certain locations that cause them to halt. However, computer models consisting of just the 'fast' and 'immobile' state could not correctly re-capture the way Ras molecules moved along the membrane. Lee et al. found that for these models to mimic the movement of Ras, a third 'intermediate' state of Ras mobility needed to be included.

To investigate this further, Lee et al. created a fluorescent map that overlaid all the individual paths taken by each Ras molecule. The map showed regions in the membrane where the Ras molecules had stopped and possibly clustered together. Each of these 'immobilization domains' were then surrounded by an 'intermediate domain' where Ras molecules had begun to slow down their movement. Although the intermediate domains did not last long, they seemed to guide Ras molecules into the immobilization domains where they could cluster together with other molecules. From there, the cell constantly removed Ras molecules from these membrane domains and returned them back to their 'fast' diffusing state.

Mutations in Ras proteins occur in around a third of all cancers, so a better understanding of their dynamics could help with future drug discovery. The methods used here could also be used to investigate the movement of other signaling molecules.

resolution imaging experiments using immuno-EM (*Prior et al., 2003*; *Plowman et al., 2005*) or quantitative superresolution microscopy (*Nan et al., 2015*; *Huang et al., 2015*) were instrumental to revealing the existence of Ras multimers, the resulting images were mostly static and provided limited information about the spatiotemporal dynamics of Ras – membrane domain interactions.

Live-cell single-particle tracking (SPT) (*Schmidt et al., 1996*; *Kusumi et al., 2005*; *Saxton and Jacobson, 1997*) complements static imaging by providing information about molecular motions, and it has been used to study Ras dynamics on the membrane (*Lommerse et al., 2006*; *Murakoshi et al., 2004*; *Lommerse et al., 2004*). The underlying rationale is that interactions of Ras with different membrane domains and signaling partners would manifest as varied diffusion behavior. Indeed, using SPT, Murakoshi et al. observed transient events of Ras immobilization on the membrane, which became more frequent upon epidermal growth factor stimulation, potentially reflecting the formation of signaling complexes or interactions with raft domains (*Murakoshi et al., 2004*). Lommerse and colleagues also used SPT to probe Ras diffusion and similarly observed transient and context-dependent confinement of Ras in membrane regions not more than 200 nm in diameter (*Lommerse et al., 2006*).

These prior studies offered important initial insight into the potential connections between Ras diffusion, function, and membrane organization, but the technical constraints of traditional SPT limited the imaging throughput and depth of analysis in these studies. Typically, only a few tens of trajectories could be obtained from each experiment, which precluded detailed and quantitative characterization of the heterogeneous and stochastic nature of molecular diffusion. In consequence, while the studies consistently reported two diffusion states – a 'free' diffusion state and another 'immobile' state, it remains to be seen whether a two-state model adequately recapitulates Ras membrane dynamics (*Lommerse et al., 2006*; *Murakoshi et al., 2004*; *Lommerse et al., 2004*). Thus, the nature of the membrane domains occupied by each of these states and how Ras molecules transition between the states in connection with multimer formation and signaling remain unclear.

Recent years have seen significant advances in both experimental (*Manley et al., 2008*; *Benke et al., 2012*; *Huang et al., 2018*; *Basu et al., 2018*; *English, 2015*; *Cutler et al., 2013*) and data analysis strategies (*Persson et al., 2013*; *Hansen et al., 2018*; *Liu et al., 2015*; *Jaqaman et al., 2008*; *Chenouard et al., 2014*; *Shen et al., 2017*; *Monnier et al., 2015*; *Ito et al., 2017*; *Lindén and Elf, 2018*) of SPT, some of which have dramatically improved the information through-put. Among others, spt-PALM combines SPT with photoactivated localization microscopy (PALM) to enable single molecule tracking under dense labeling conditions through stochastic photoswitching (*Manley et al., 2008*). With spt-PALM, it is routine to acquire thousands of diffusion trajectories from a single cell. A growing list of software tools has also been developed to facilitate spt-PALM data analysis (*Persson et al., 2013*; *Hansen et al., 2018*; *Jaqaman et al., 2008*; *Ito et al., 2017*; *Newby et al., 2018*). For example, variational Bayes SPT (vbSPT) allows construction of a detailed diffusion model from spt-PALM data with parameters such as the number of states, the diffusion coefficient and the occupancy of each state, as well as the state transition rates even when the individual trajectories are short (*Persson et al., 2013*). Additional methods have also been introduced to quantitate various aspects of diffusion dynamics from SPT trajectories (*Chenouard et al., 2014*; *Ito et al., 2017*; *Manzo and Garcia-Parajo, 2015*). These advances help overcome the limitations of conventional SPT and make it possible to analyze Ras membrane dynamics in much greater depth.

Here, we report our efforts on combining spt-PALM with detailed trajectory analysis to reveal previously unknown aspects of Ras diffusion on the cell membrane. With carefully controlled expression levels and photoactivation rate, spt-PALM trajectories of PAmCherry1-tagged KRas$^{G12D}$ (KRas with an activating mutation and thus primarily GTP-bound) consistently reported three diffusion states, including a fast diffusion state, an immobile state, and a previously unidentified diffusion state with intermediate mobility. Leveraging the large number of trajectories, we were able to spatially map the diffusion states to distinctive membrane domains, estimate the size and lifetime of each domain, and define the spatial relationship between the domains. Moreover, in analyzing how KRas$^{G12D}$ transitions from one diffusion state to another, we discovered that KRas$^{G12D}$ diffusion follows a non-equilibrium steady state (NESS) model with net mass flow from the fast state to the immobile state, potentially coupled to the endocytic trafficking and membrane recycling of KRas$^{G12D}$. Based on these results, we propose a new model to describe the membrane dynamics of KRas$^{G12D}$, where nested membrane nanodomains dictate the diffusion and trafficking, with implications in Ras multimer formation and signaling.

## Results

### KRas$^{G12D}$ diffuses on the membrane in three distinct states

To investigate the lateral diffusion properties of KRas$^{G12D}$ under controlled expression levels, we established isogenic U2OS cells stably expressing PAmCherry1-KRas$^{G12D}$ under doxycycline (Dox) regulation (*Nan et al., 2015*). The expression level of PAmCherry1-KRas$^{G12D}$ could be tuned from a level below that of the endogenous KRas at <1 ng/mL Dox to highly over-expressed at 5–10 ng/mL Dox (*Figure 1A*). Initially data were collected from cells expressing KRas$^{G12D}$ at a moderate level by inducing at 2 ng/mL Dox. The photoactivatable fluorescent protein PAmCherry1 has been widely used for quantitative PALM and spt-PALM (*Subach et al., 2009*). Owing to the good single-molecule brightness of activated PAmCherry1, we were able to track individual PAmCherry1-KRas$^{G12D}$ molecules at frame rates up to ~83 Hz (i.e.,~12 ms/frame) with a low excitation dose (~400 W/cm$^2$ at 561 nm). The low spontaneous photoactivation rate of PAmCherry1 also permits clean single-molecule imaging even at high expression levels, yielding as many as hundreds of thousands of trajectories per cell via spt-PALM (*Figure 1B* and *Video 1*). Under these conditions, the average trajectory lengths were ~4 and~5 frames for data acquired at 12 ms/frame and 35 ms/frame rates, respectively (*Figure 1—figure supplement 1*). Despite the faster frame rate, data acquired at 12 ms/frame had a lower signal-to-noise ratio, causing a more frequent loss of molecules during tracking to yield significantly shorter trajectories (~50 ms average duration) than data acquired at 35 ms/frame (~175 ms average duration). We therefore used both frame rates in this work for the benefit of better temporal resolution or spatial precision.

A close inspection of the individual trajectories clearly shows larger diffusive steps intermittent with moments of transient entrapment, indicating the presence of multiple diffusion states and

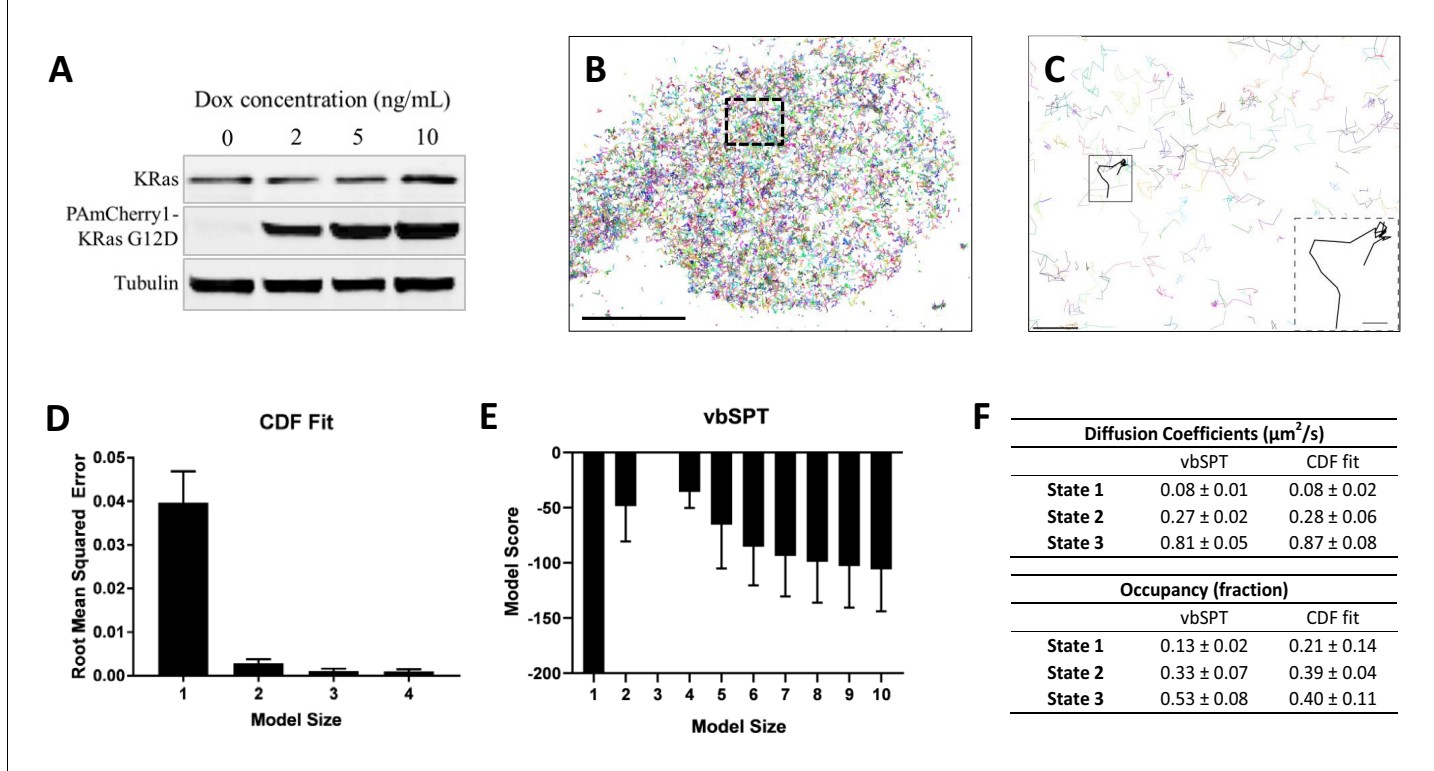

**Figure 1.** Defining the membrane diffusion model of KRas[G12D] using spt-PALM and vbSPT. (**A**) Western blot showing the increasing expression level of PAmCherry1-KRas[G12D] with increasing doxycycline (Dox) concentration; (**B**) Example trajectory map of membrane KRas[G12D] acquired at 12 ms frame rate using total internal reflection fluorescence (TIRF) illumination. Each line represents an individual Ras molecule coordinate over time acquired for the duration of the movie (20 min). Only a subset of all trajectories is plotted. Scale bar: 10 µm; (**C**) Expanded view of the boxed region in B). Only a subset of all of the trajectories in the boxed region is shown to allow unhindered view of individual Ras trajectories. Inset shows a KRas[G12D] trajectory displaying multiple diffusion states. Scale bars: main figure: 1 µm; inset: 200 nm; (**D**) Determining the optimal model size for KRas[G12D] membrane diffusion using CDF fitting, with smaller root mean squared error indicating a better model (n = 8); (**E**) Determining the optimal model size for KRas[G12D] membrane diffusion using vbSPT, with smaller absolute model score (i.e., score of zero being the best global model) indicating a better model (n = 5); (**F**) Comparing the model parameters obtained from CDF fit and vbSPT, both using a three-state model for KRas[G12D] membrane diffusion. State transition probabilities were not inferred from CDF fit and therefore not included in the comparison. Error bars are 95% confidence intervals (CIs). The online version of this article includes the following source data and figure supplement(s) for figure 1:

**Source data 1.** Excel sheet for data used for generating panels D, E, and F.
**Figure supplement 1.** Trajectory length histograms.
**Figure supplement 2.** Impact of particle density on diffusion model reconstruction.
**Figure supplement 3.** Impact of trajectory connection distance on vbSPT model output.
**Figure supplement 4.** Impact of localization error on vbSPT model output.
**Figure supplement 5.** vbSPT model output on experimental spt-PALM datasets acquired at high particle densities.
**Figure supplement 6.** Photon yield and localization accuracy at the different frame rates used in this work.

frequent state transitions (*Figure 1C* and inset). Similar observations were reported for both HRas and KRas in previous low throughput SPT experiments, where two diffusion states – a 'fast' state and an 'immobile' diffusion state – were detected (*Lommerse et al., 2006*; *Murakoshi et al., 2004*).

Since spt-PALM offers a much larger number of trajectories, we first asked whether KRas[G12D] diffusion on the cell membrane could indeed be described by a simple two-state model. To this end, we used two methods to analyze the Ras diffusion trajectories. The first approach fits cumulative distribution function (CDF) for Brownian motion to the squared displacements of Ras trajectories to extract diffusion coefficients and the respective occupancies of the diffusion states (*Schütz et al., 1997*). The second method, vbSPT, treats particle diffusion and the associated state transitions with a Hidden Markov Model and performs model selection through variational inference (*Persson et al., 2013*). Of note, vbSPT is well suited for analyzing large numbers of short trajectories such as those obtained via spt-PALM.

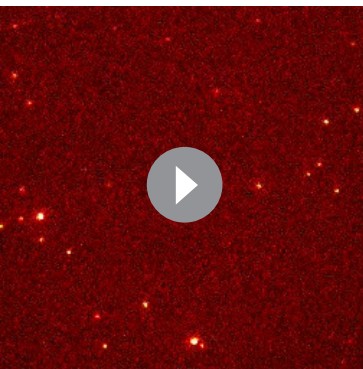

**Video 1.** Clip of a raw spt-PALM video showing PAmCherry1-KRas[G12D] diffusion on U2OS cell membrane. Data acquired at 35 ms/frame. Cells were induced with 5 ng/mL doxycycline before imaging. https://elifesciences.org/articles/46393#video1

We found that particle densities higher than 0.03 $\mu m^{-2}$ per frame under our experimental conditions (12 ms/frame with the fastest diffusion rate at ~1 $\mu m^2$/s) led to occasional misconnected trajectories, and that even a small fraction of such misconnected trajectories could lead to skewed model outputs with vbSPT (*Figure 1—figure supplement 2*). In addition, the threshold for maximum particle displacement between adjacent frames also had an impact on trajectory misconnection, although to a lesser extent as tested with simulated trajectories (*Figure 1—figure supplements 2* and *3*). Thus, for diffusion model construction, we chose to use a high frame rate (12 ms/frame) and a low particle density (<0.03 $\mu m^{-2}$) to eliminate misconnected trajectories while maintaining a sufficient number of trajectories.

Using trajectories acquired and analyzed with the above precautions, both CDF fitting and vbSPT yielded similar three-state models for KRas[G12D] diffusion on the membrane of live U2OS cells. Specifically, CDF fitting to a three-state model had significantly lower residual error compared to a single- or a two-state model and further increasing the model size did not decrease the error (*Figure 1D*), indicating that a three-state model is sufficient to describe the data. For vbSPT, a score equal to zero indicates the best model, a condition that was met with a three-state model but not with larger or smaller size models (*Figure 1E*). To rule out the possibility of imprecise single-molecule localizations causing vbSPT to misinterpret two-state spt-PALM data as three states, we performed vbSPT analysis on simulated trajectories based on a two-state model with varying localizations errors (0, 20, 40, 80, 100 nm) added. Even at the highest localization errors (100 nm), vbSPT correctly retrieved a two-state model (*Figure 1—figure supplement 4*) from the simulated data, suggesting that the three-state model derived from experimental spt-PALM data is unlikely a result of localization errors in SPT.

The diffusion coefficient and the occupancy for each of the diffusion states were in good agreement between the two analysis methods and within each method when applied to different cells under the same conditions, as evidenced by the small errors (*Figure 1F*). Datasets with high particle densities can return models with different sizes, sometimes also with aberrant model parameters (*Figure 1—figure supplement 2B–D* and *Figure 1—figure supplement 5A–B*); even so, the histogram of all vbSPT-derived diffusion coefficients still showed three distinct clusters (*Figure 1—figure supplement 3C*) corresponding to the three states listed in *Figure 1F*. Thus, we concluded that the membrane diffusion of KRas[G12D] under our experimental conditions is best described by a three-state model, demonstrating the existence of an intermediate state not detected in previous studies. Between the two methods, vbSPT was used for most subsequent analyses in the remainder of this work because it infers the transition probabilities and state identities for every time step whereas CDF does not.

The diffusion coefficient of the slowest state in *Figure 1F* is comparable to that expected from single-molecule localization error (~40 nm, *Figure 1—figure supplement 6*), which implied that the actual diffusion of KRas[G12D] in this state may be even slower than it appeared. To test this hypothesis, we acquired spt-PALM data at a slower frame rate (35 ms/frame) to improve the localization accuracy of slowly moving molecules since more photons could now be collected for each PAmCherry1 molecule in a single frame (*Figure 1—figure supplement 6*). Indeed, these datasets reported a significantly smaller diffusion coefficient (0.02 $\mu m^2$/s) for the slowest state than that obtained earlier (0.08 $\mu m^2$/s) using data taken at 12 ms/frame. This result suggests that the slowest diffusion state of KRas[G12D] is essentially an immobile state, consistent with previous reports (*Lommerse et al., 2006*; *Murakoshi et al., 2004*).

# KRas$^{G12D}$ diffusion states correspond to distinct membrane domains

The diffusion model presented in *Figure 2A* summarizes the results from the spt-PALM trajectory analyses using vbSPT. Each circle represents one of the diffusion states with arrows indicating the transition probabilities between pairs of states. A notable feature of this model is that there appears to be a defined state transition path: KRas$^{G12D}$ molecules always transition between the fast (F) and the immobile (I) states by going through the intermediate (N) state, and direct transitions between the fast and the immobile states almost never occur. In order to confirm this transition path, we compared the distribution of step sizes relative to the immobile state steps, since different step sizes reflect different diffusion coefficients. Consistent with the state transition path observed in *Figure 2A*, the histogram of step sizes immediately adjacent to the immobile steps corresponded to the intermediate diffusion state (*Figure 2B*, blue) while the distribution of the remaining steps had a broader peak implying a mixture of both fast and intermediate diffusion steps (*Figure 2B*, where the black color indicates a mixture of states). As expected, the step sizes assigned to the immobile states (*Figure 2B*, red) are even smaller compared to that of the other two states. The clear separation of these three step size distributions confirms the above-mentioned transition path through the intermediate state. The distinctions in step sizes among the three states were even more obvious on data taken at 35 ms/frame, which had better single-molecule localization precision (*Figure 2—figure supplement 1*). Thus, the intermediate state is not merely a state with intermediate mobility but effectively an obligatory link between the immobile and the fast states of KRas$^{G12D}$.

The observed state transition path may arise from at least two potential scenarios. In the first scenario, fast diffusing KRas$^{G12D}$ may transition into the intermediate then the immobile state through spontaneous conformational changes unrelated to slow or static membrane structures. Alternatively, the immobile states could be caused by KRas$^{G12D}$ transiently binding to stationary molecules or structures (termed 'immobilization sites' or 'immobilization domains') residing in membrane regions (referred to as 'intermediate domains') that confer intermediate mobility to KRas$^{G12D}$. Consequently, these intermediate domains would act as transition zones between membrane regions where KRas$^{G12D}$ exhibits fast diffusion and the sites of KRas$^{G12D}$ immobilization, yielding the observed state transition path. In either case, the intermediate and the immobile states of KRas$^{G12D}$ would be temporally and spatially correlated. It is only in the latter case, however, that we would observe multiple visits to the same intermediate or immobilization domains by different KRas$^{G12D}$ molecules, provided that both domains have lifetimes longer than our temporal resolution. Of note, even in the second scenario, KRas$^{G12D}$ targeting to the intermediate or the immobilization domains may be accompanied by changes in conformation.

To distinguish between the two scenarios, we performed auto- and cross-correlation analysis on the locations of KRas$^{G12D}$ exhibiting a certain diffusion state (referred to hereafter as state coordinates). We first visually examined the spatial distributions of the states by slicing each raw image stack into one-minute time substacks and plotting the state coordinates on the same map, with each color representing one of the states (*Figure 2C*, *Figure 2—figure supplement 2*, and *Video 2*). Each diffusion trajectory typically contributes only a few points to the plots as limited by its short duration, and the points from multiple trajectories accumulate over time (up to 1 min in this case) to 'paint' a map of the membrane regions associated with each diffusion state. Despite yielding relatively short trajectories, the rapid turn-over of PAmCherry1 allowed more efficient sampling ('painting') of the membrane domains by KRas$^{G12D}$ in the field of view. As shown in *Figure 2C*, the intermediate state locations and the immobile state locations not only co-clustered, but also each appeared to self-cluster. Specifically, regions corresponding to the intermediate states (blue) often connect to give rise to nanoscopic domains a few hundred nm in size, and the vast majority of the immobilization sites (red) are surrounded by the intermediate domains. By contrast, regions corresponding to the fast state occupy the majority of the membrane area. While both the intermediate and the immobilization domains appeared to be dynamic, a time-lapse domain map (*Video 2*) showed that at least some of these domains could last a few minutes (to be further addressed below in *Figure 3*). Thus, spatial mapping of the KRas$^{G12D}$ state coordinates provided visual evidence for the physical presence of nested, nanoscopic domains conferring the distinct KRas$^{G12D}$ diffusion states.

We next used pair correlation function ($g(r)$) to quantitate the spatial relationship between the KRas$^{G12D}$ diffusion states (*Figure 2D–E*). The function $g(r)$ measures the ratio of the number of

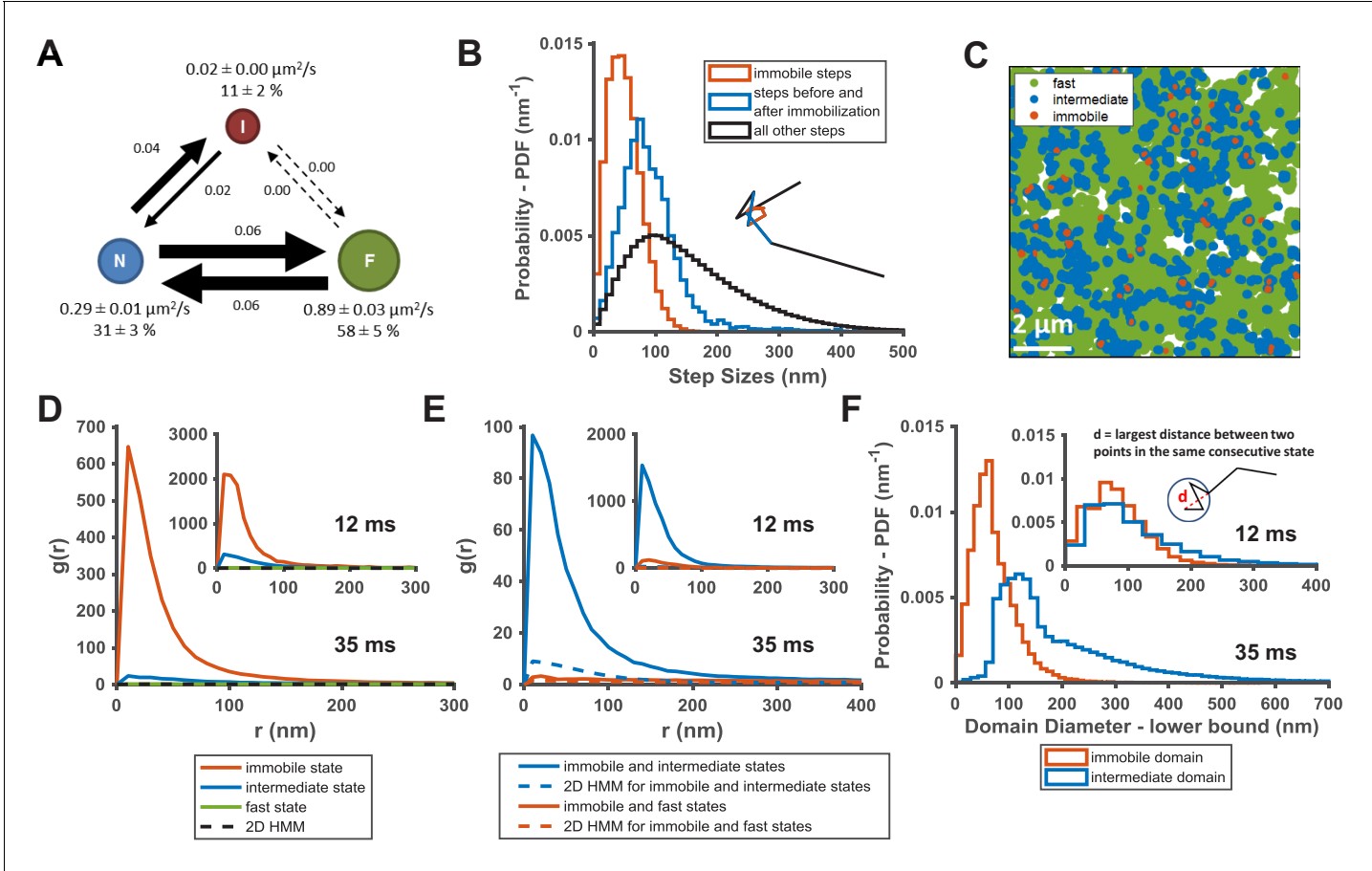

**Figure 2.** KRas[G12D] diffusion states are associated with distinct membrane domains. (**A**) The three-state model for KRas[G12D] diffusion with F, N, and I, representing the fast, the intermediate, and the immobile states, respectively. Model parameters were inferred using vbSPT on spt-PALM datasets with at least 30,000 trajectories obtained on cells induced with 2 ng/mL Dox. The arrows indicate state transitions (i.e. the probability of switching to a different state in the next frame) and the area of the circle and the thickness of the arrows are both roughly scaled to reflect their relative values. All parameters were derived from data acquired at 12 ms frame interval except for the diffusion coefficient of the immobile state, which was inferred from data taken at 35 ms frame interval. Error bars are 95% CIs (n = 8); (**B**) Step size histograms for immobilization events (red), one step before or after the immobilization event (blue), and all other steps (black). A diffusion step was part of an immobilization event if immobile state was assigned to that trajectory segment by vbSPT (n = 14, see Materials and methods); (**C**) Map of the membrane locations where KRas[G12D] molecules exhibit specific diffusion states (referred to as state coordinates) within a one-minute duration (taken from a spt-PALM dataset of ~20 min total duration). Red, blue, and green dots represent locations of the immobile, the intermediate, and the fast states, respectively, with each rendered circles scaled proportionally to the mean diffusion coefficient for the state; (**D**) Pair correlation analysis on the averaged state coordinates across multiple, one-minute segments of longer spt-PALM datasets. The same color coding as in B) was used to distinguish the three states. For this analysis, molecules in the same diffusion state in successive frames only contributed a single, averaged state coordinate. The average state coordinates of all molecules captured within a one-minute segment were used for correlation analysis, and the results from multiple one-minute segments were averaged to yield the plot. The negative control was generated through a 2D Markovian simulation, and the resulting trajectories were analyzed the same as the experiment (see Materials and methods); (**E**) Cross correlation analysis between pairs of diffusion states. The state coordinates were processed the same way as in D) prior to the correlation analysis, except that the correlation was performed between two different diffusion states. The negative control was generated through a 2D Markovian simulation, and the resulting trajectories were analyzed the same as the experiment (see Materials and methods); (**F**) Estimating the lower bound size for the immobile and the intermediate domains. The estimation was based on the maximum distance traveled by the molecule while in the same diffusion state. *D-F) The main panel shows results inferred from data taken at 35 ms frame intervals for improved localization precision. The inset shows the data taken at 12 ms/frame (n = 14 for 12 ms and n = 7 for 35 ms datasets).

The online version of this article includes the following source data and figure supplement(s) for figure 2:

**Source data 1.** Excel sheet for data used for generating panels B, D, E, and F.
**Figure supplement 1.** Spatial analysis of KRas[G12D] membrane domain properties using data acquired at 35 ms per frame.
**Figure supplement 2.** Temporal evolution of the membrane domains associated with each KRas[G12D] diffusive state.

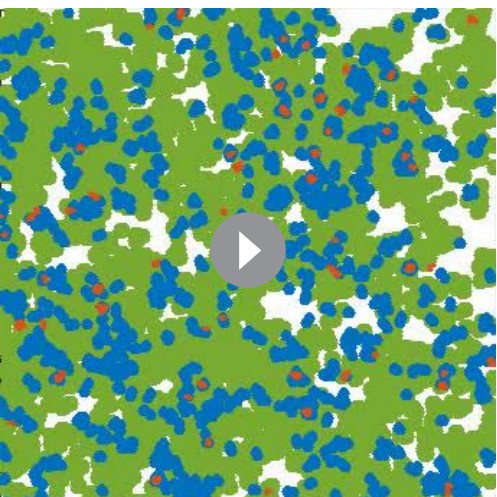

**Video 2.** Time-lapse (1 min/frame) video of the domain map calculated from individual trajectories within each substack. Red: immobilization domains (sites); blue: intermediate domains; green: fast (free) domains. The image area is around 10 × 10 μm².
https://elifesciences.org/articles/46393#video2

particles located a distance ($r$) from a given particle to that expected from a complete spatial randomness (see Materials and methods). Here, the $g(r)$ could be calculated for particles in the same diffusion state (auto-correlation) or between two different diffusion states (cross-correlation); in either case, amplitudes of $g(r)$ significantly greater than that expected for a random distribution indicate spatial clustering. When multiple KRas$^{G12D}$ molecules visit the same domain, each at a different time point but exhibiting the same diffusion state, $g(r)$ would detect spatial auto-correlation for the given state. To avoid false clustering due to the same molecule staying in the same state across multiple frames, we used the averaged state coordinate for each continuous trajectory segment that stayed in the same state for more than two consecutive time points (see Materials and methods). Results from both datasets taken at 35 ms/frame (main panels) and those at 12 ms/frame (inset) are shown for comparison (*Figure 2D–E*).

Consistent with the visual observation earlier (*Figure 2C*), coordinates of the immobile and the intermediate states each showed significant clustering in the $g(r)$ plots averaged across each 1 min raw image stacks, whereas $g(r)$ of the fast state was barely above random across the full range of $r$ analyzed (*Figure 2D*). All $g(r)$ negative controls were generated with a 2D Markovian simulation of diffusing particles with no associated domains (see Materials and methods), and the simulated trajectories were processed through the same analysis pipeline as the experimental data. As expected, the averaged state coordinates of the simulated negative control had values close to one and showed no peak in the $g(r)$ plots. Furthermore, $g(r)$ based on spatial cross-correlation analysis clearly indicated co-clustering between the immobile and the intermediate state positions but not with the fast diffusion state (*Figure 2E*).

## Transient nanodomains mediate the intermediate and the immobile states of KRas$^{G12D}$

We also estimated the lower-bound size of the domains associated with the immobile and the intermediate states of KRas$^{G12D}$ by calculating the maximum distance a molecule traveled while in a domain (i.e., longest distance between two points within consecutive steps taken while in the same state). Shown in the main panel of *Figure 2F* are the histograms of the estimated domain sizes determined from data taken at 35 ms/frame, based on which we determined that the mean diameters of the intermediate and the immobile membrane domains were approximately ~200 nm and ~70 nm, respectively. This is consistent with the notion that most immobilization domains are likely surrounded by intermediate domains. The distinction between the two domains became much less significant with data taken at 12 ms/frame (*Figure 2F*, inset), which we attributed to the shorter trajectory durations (~50 ms at 12 ms/frame compared to ~175 ms at 35 ms/frame; see *Figure 1—figure supplement 1*), which in turn was due to the lower photon yield per frame from single PAm-Cherry1 molecules at this fast frame rate. In essence, the molecules failed to sample a large enough area within the short duration of the trajectories to report the domain size accurately. In addition, the distribution of the minimum intermediate domain size appeared to have at least two peaks at ~120 nm and ~230 nm, implying that there may potentially be multiple types of intermediate domains (*Figure 2F*).

To understand the temporal behavior of the immobilization and the intermediate domains associated with KRas$^{G12D}$, we extended $g(r)$ calculations as in *Figure 2* from one minute to longer time intervals. The rationale was that, as the time interval for calculating $g(r)$ increases beyond the lifetime

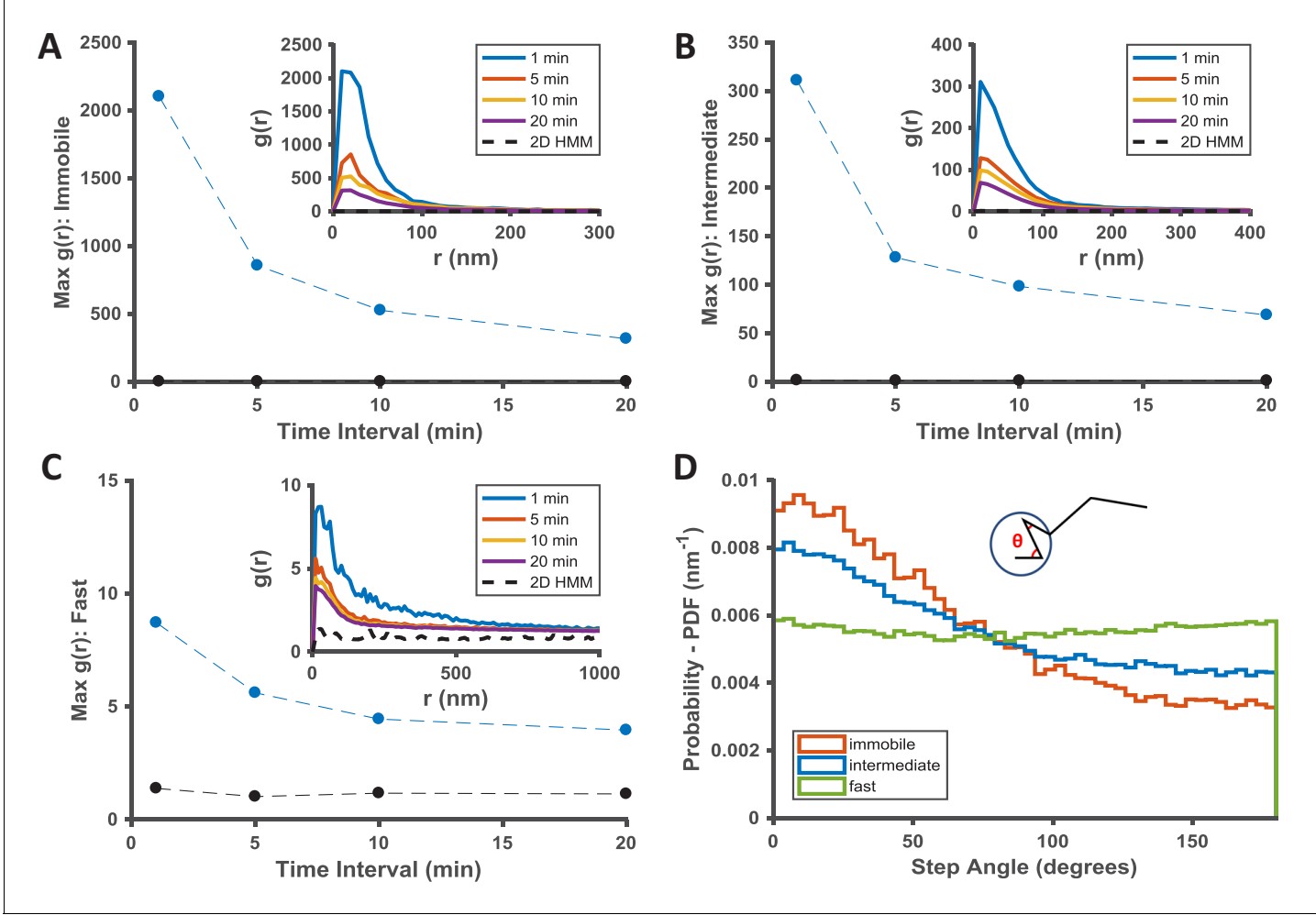

**Figure 3.** Temporal properties of the KRas[G12D]-associated immobile and intermediate domains. (A–C) Pair correlation analysis of the state coordinates at different time intervals (1, 5, 10, and 20 min). The amplitude (maximal $g(r)$ value) of the pair correlation function at each time interval was plotted in the main panel with the raw pair correlation plots shown in the inset. A-C show pair correlation functions of averaged coordinates for the immobile, the intermediate, and the fast states, respectively (see Materials and methods). The negative control in each case was generated through a 2D Markovian simulation, and the resulting trajectories were analyzed the same as the experiment (see Materials and methods); D) Deflection angle analysis on KRas[G12D] diffusion trajectories separated by diffusion states (red: immobile; blue: intermediate; green: fast). The deflection angle was calculated as the angle between two successive segments of the trajectory while the molecule was in the same diffusion state. *Results shown for data acquired at 12 ms/frame (n = 14).

The online version of this article includes the following source data for figure 3:

**Source data 1.** Excel sheet for data used for generating all the panels.

of a domain, the chance of observing KRas[G12D] molecules visiting the same domain (i.e., exhibiting the same diffusion state in close proximity) should decrease, resulting in lower $g(r)$ amplitudes. Indeed, as shown in *Figure 3A–C*, for dataset acquired at 12 ms frame interval, the peak amplitudes of $g(r)$ for both the immobile (*Figure 3A*) and the intermediate (*Figure 3B*) states decreased significantly after ~5 min with further decay at increasing time intervals, indicative of finite lifetimes for both nanodomains, likely on the order of minutes on average (see also *Figure 2—figure supplement 1* for results with data taken at 35 ms/frame). For the limited temporal resolution of this analysis, we likely only detected relatively stable domains with lifetimes longer than 1 min, and the presence of more transient intermediate or immobilization domains should not be ruled out.

To gain insight into how KRas[G12D] interacts with the different membrane domains, we also analyzed the frame-to-frame deflection angle for the molecules within each domain. The deflection angle measures the relationship between the current and the preceding step: a complete random

walk would yield a flat distribution of deflection angles, whereas a preference for acute angles indicates more 'returning' steps. The measurement will likely be affected by localization error: for individual angles, larger localization error (relative to the step sizes) would add significant noise to the measured angles; for ensemble measurement of a large number of angles, however, the localization error would affect all angles in an unbiased manner. Thus, despite the finite localization precision at our frame rates, we expect that the measured step angles to reflect the interactions between KRas$^{G12D}$ and the membrane domains. Indeed, as shown in *Figure 3D*, KRas$^{G12D}$ molecules trapped in either the immobilization or the intermediate domains (the red and the blue lines) were more likely to exhibit acute deflection angles, potentially due to backward movements at the domain boundaries. Between the two domains, the enrichment of acute angles was more significant for the immobile state because the associated domains were smaller, such that KRas$^{G12D}$ molecules had a higher chance of hitting the domain boundaries. In comparison, KRas$^{G12D}$ molecules in the fast state exhibit (*Figure 3D*, the green line) equal probabilities of moving in all directions, consistent with Brownian motion.

## KRas$^{G12D}$ is constitutively depleted from the immobile state and replenished to the fast state

The small variance in the estimated model parameters from data taken on different cells, be it from the same or different samples (*Figure 2A*), led us to hypothesize that KRas$^{G12D}$ membrane diffusion is in a steady state. To verify this, we divided each spt-PALM dataset with a minimum of 40,000 trajectories into four quarters (each with ~10,000 trajectories and typically ~5 min long) and computed the diffusion model for each quarter using vbSPT. As *Figure 4A* shows, the model parameters for all four quarters were essentially identical, which is the case for all qualifying datasets, confirming that KRas$^{G12D}$ diffusion is indeed in a steady state, at least in U2OS cells and at the investigated time scales (up to ~20 min).

In contradiction to KRas$^{G12D}$ diffusion being in a steady state, however, we found that the diffusion model as presented in *Figure 2A* cannot self-sustain. When using experimentally derived model parameters to simulate how the three-state system evolves over time (see Materials and methods), we observed that the system quickly deviated from its initial configuration and instead stabilized at an entirely different set of state occupancies (*Figure 4B*). In the new, 'equilibrated' system configuration, KRas$^{G12D}$ spends as much as ~50% of its time in the immobile state, significantly more than the observed steady state occupancy of ~11%. The fast state is the opposite: the population residing in this state is significantly reduced from ~58% to ~25%. By contrast, the intermediate state changes only slightly (~31% vs ~24% for the experimental observations and the simulations, respectively). We confirmed that the simulated equilibrium probabilities were consistent with the principle of detailed balance (*Zuckerman, 2010*) (*Figure 4C*). Next, we also verified that the experimentally determined state occupancies in *Figure 2A* were not an artifact of vbSPT, since vbSPT correctly retrieved the steady state model parameters when applied to simulated trajectories from steady state models with varying input parameters (*Figure 4—figure supplement 1*). Therefore, we concluded that the model in *Figure 2A* represents a non-equilibrium steady state (NESS).

To further characterize the NESS, we calculated the mass flow for each of the three KRas$^{G12D}$ diffusion states as the change in state occupancy per time interval. A positive net flow rate or a ratio of in- vs out-flux greater than one indicates an accumulation of mass for the state, while a negative flow rate or a ratio of flux less than one indicates the opposite. As shown in *Figure 5A and B*, within the NESS there is a net influx of KRas$^{G12D}$ molecules to the immobile state and a net outflux of molecules from the fast state, whereas the in- and out-fluxes for the intermediate state are comparable. We also calculated the mass flow for each of the three arms in the diffusion model in *Figure 2A* – in the clockwise direction, it would be the flow from the fast state to the intermediate state (F to N), intermediate to immobile (N to I), and immobile to fast (I to F). The results of this calculation are shown in *Figure 5C*, where a positive value in the y axis (net mass flow between a pair of states) indicates mass flow in the designated direction, and a negative value indicates flow in the opposite direction. Consistent with results in *Figure 5A and B*, the dominant net mass flow through the NESS is unidirectional – from the fast state to the intermediate to the immobile state (*Figure 5C*) – with minimal 'leakage' from the fast to the immobile state.

These results are consistent with the simulated system relaxation to equilibrium shown in *Figure 4B*, where the immobile and the fast diffusion states changed occupancies the most. For the

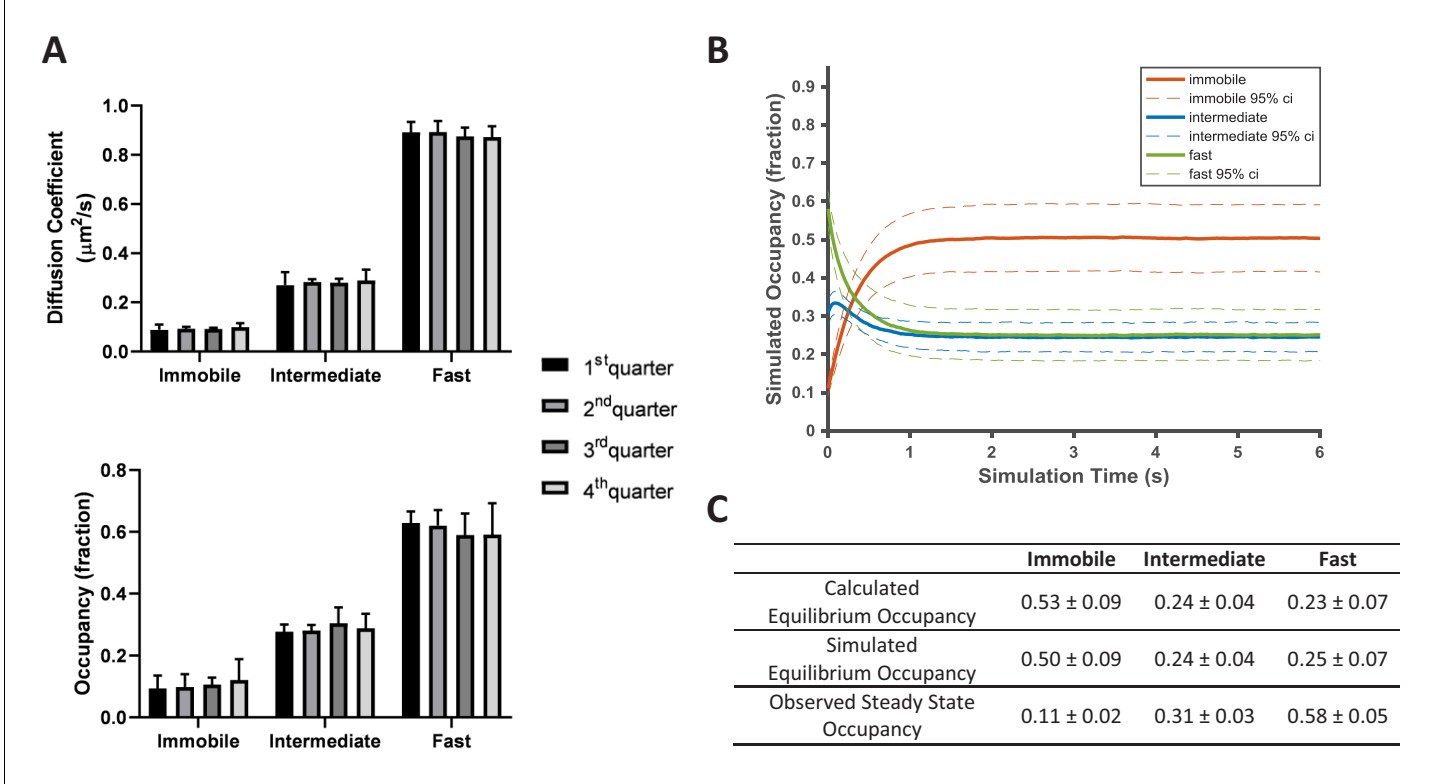

**Figure 4.** KRas[G12D] diffusion on the cell membrane is in a non-equilibrium steady state. (**A**) Time invariance of the KRas[G12D] diffusion model. A single ~20 min spt-PALM dataset was segmented into four quarters with each quarter containing ~10,000 trajectories (in ~5 mins), each analyzed separately using vbSPT to obtain the model parameters such as the diffusion coefficients (upper panel) and the state occupancies (lower panel). Results from multiple spt-PALM datasets were grouped and plotted (n = 4); **B**) Temporal evolution of the KRas[G12D] diffusion model in simulated runs. The system was setup according to the experimental model parameters (number of states, state occupancies, diffusion coefficients, and state transition rates) as shown in *Figure 2A*. The system was then allowed to evolve based on the input, with the new state occupancies recorded every time step (12 ms) and plotted (see Materials and methods). Similar to *Figure 2A*, only movies with minimum of 30,000 trajectories were simulated (n = 8); **C**) Table summarizing the calculated, simulated, and experimentally observed occupancies for each of the states. *All error represents 95% CIs.

The online version of this article includes the following source data and figure supplement(s) for figure 4:

**Source data 1.** Excel sheet for data used for generating all the panels.

**Figure supplement 1.** Validating vbSPT output accuracy on simulated trajectories using different model parameter inputs.

KRas[G12D] NESS system to be sustained over time as we observed experimentally, KRas[G12D] would need to be replenished into the fast diffusion state and removed from the immobile state. Indeed, KRas[G12D] has previously been shown to undergo a constant exchange between the plasma membrane and the cytosol, and internalized KRas[G12D] is collected at recycling endosomes and transported back to the plasma membrane (*Schmick et al., 2014*; *Schmick et al., 2015*). Our analyses suggest that the loss of KRas[G12D] from the membrane could be through the immobile state, and the replenishment through the fast state. At present, it is unclear whether the intermediate state has no exchange with the cytosol or has active exchange with equal gain and loss. Accordingly, the membrane trafficking of KRas[G12D] should follow the model presented in *Figure 5D*, where the arrows indicate the net mass flow between the connected states as well as between the states (F or I) and the environment (such as the cytosol).

## KRas[G12D] diffusion model is invariant over a range of expression levels

Next, we sought to investigate whether experimental conditions such as expression level would alter the diffusion properties of KRas[G12D]. An important observation on Ras nanocluster (multimer) formation is that the fraction of clustered molecules remains constant over a broad range of expression levels (*Plowman et al., 2005*). This unusual property has led to two hypothetical mechanisms of

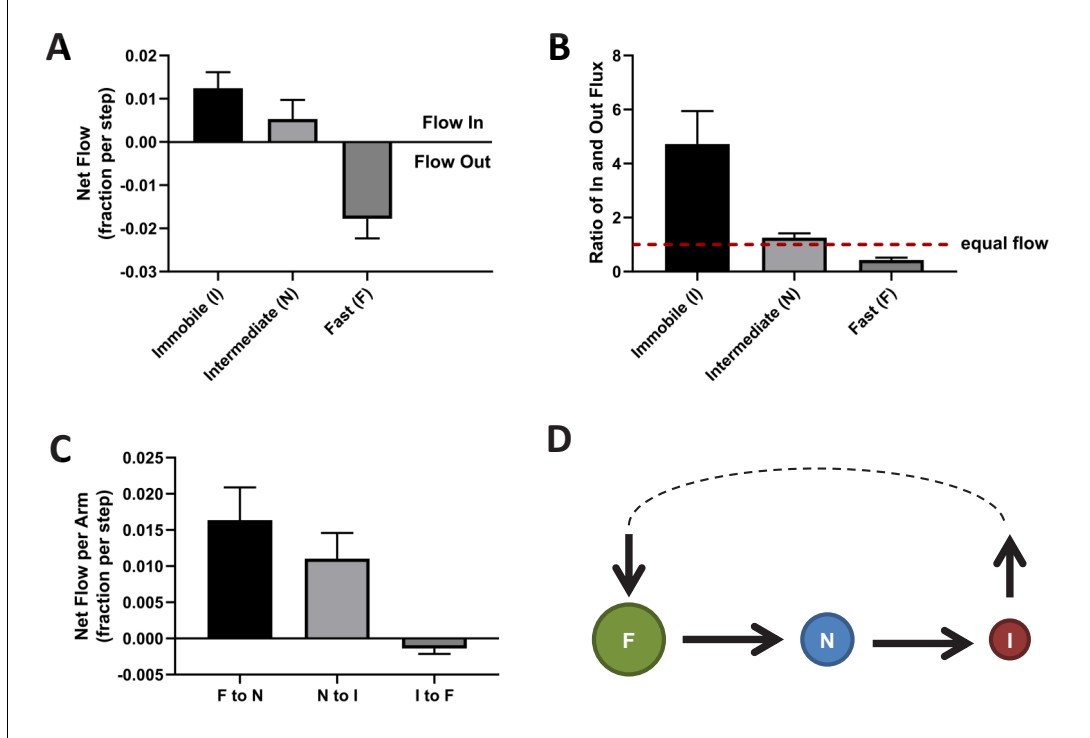

**Figure 5.** Directional mass flow between KRas[G12D] diffusion states. (**A**) Net mass flow per state, defined as the difference between the influx (positive) and the outflux (negative) for each state and expressed as the fraction (of total KRas[G12D] population) entering (positive, flow in) or leaving (flow out, negative) the state per time interval; (**B**) Ratio of in- and outflux for each state. A ratio of one (dashed line) represents equal in- and outflux for the state, greater than one represents more influx than outflux, and less than one represents net outflux of mass from the state; (**C**) Net mass flow per arm (pair of states) in the KRas[G12D] diffusion model (**Figure 2A**). F to N and N to I are not significantly different. The states were ordered in a clock-wise direction, and the net mass flow in the direction was calculated as the difference between forward and backward mass flows, with a positive value indicating net flow in the indicated direction and a negative value the opposite direction; (**D**) Model for KRas[G12D] trafficking between the diffusion states and between the membrane system and the environment (cytosol). Arrows indicate the directional mass flow, and the dashed line represents unknown mechanisms connecting the fast and the immobile states. *All error bars are 95% CIs (n = 22).

The online version of this article includes the following source data for figure 5:

**Source data 1.** Excel sheet for data used for generating panels A, B, and C.

membrane nanocluster formation: one based on protein self-nucleation (**Plowman et al., 2005**) and another involving actomyosin activity (**Gowrishankar et al., 2012**). These active mechanisms are in contrast to passive localization of Ras to existing membrane nanodomains (e.g. via diffusion), which was thought to result in concentration-dependent multimer formation and therefore be inconsistent with the constant fraction of clustered Ras. To date, it remains controversial as to which mechanism mediates Ras multimer formation, including the basic question of whether membrane nanodomains are involved. We reasoned that, if KRas[G12D] multimers form in membrane nanodomains – for example the intermediate and/or the immobilization domains in this case – then the observed fraction(s) of KRas[G12D] in either or both the intermediate and the immobile states should also be independent of expression level, as for the fraction of Ras molecules in multimers (clusters).

To address this question, we induced PAmCherry1-KRas[G12D] at a range of expression levels using different Dox concentrations (**Figure 1A**). Similar to our previous report (**Nan et al., 2015**), the expression level of PAmCherry1-KRas[G12D] responded well to varying Dox concentrations in the isogenic cells used in this study, with the protein expression at 0 ng/mL being extremely low (only due to occasional leakage in tetR suppression) and that at 10 ng/mL about 5–10 fold higher than endogenous KRas[G12D]. When measured in terms of protein density at the membrane, the tuning range corresponds to <10 molecules per μm² at 0 ng/mL Dox to >300 molecules per μm² at 10 ng/mL Dox.

By comparing estimated model parameters using spt-PALM data of PAmCherry1-KRas[G12D] at different Dox concentrations, we found that KRas[G12D] diffusion properties remained essentially the

same across the range of expression levels investigated (*Figure 6A–B* and *Figure 6—figure supplement 1*). This model invariance is reflected across all conditions: not only was a three-state model optimal for describing the diffusion of KRas[G12D] as judged with vbSPT (not shown) and with CDF (*Figure 6—figure supplement 2*), but the diffusion coefficients of each state, the state occupancies, as well as the transition probabilities between each pair of states, were indistinguishable within the error bars.

As expected, the net mass flow rates (expressed as the change in state occupancy per time interval) of KRas[G12D] within the system also remained the same across all the Dox concentrations (*Figure 6C–D*). A similar observation was made when we acquired the trajectories at 35 ms/frame (*Figure 6—figure supplement 3*). Thus, we concluded that KRas[G12D] diffusion and trafficking on the membrane remains constant over the range of tested KRas[G12D] expression levels. Equivalently, the partitioning of KRas[G12D] in each of the three diffusive states – and the corresponding membrane

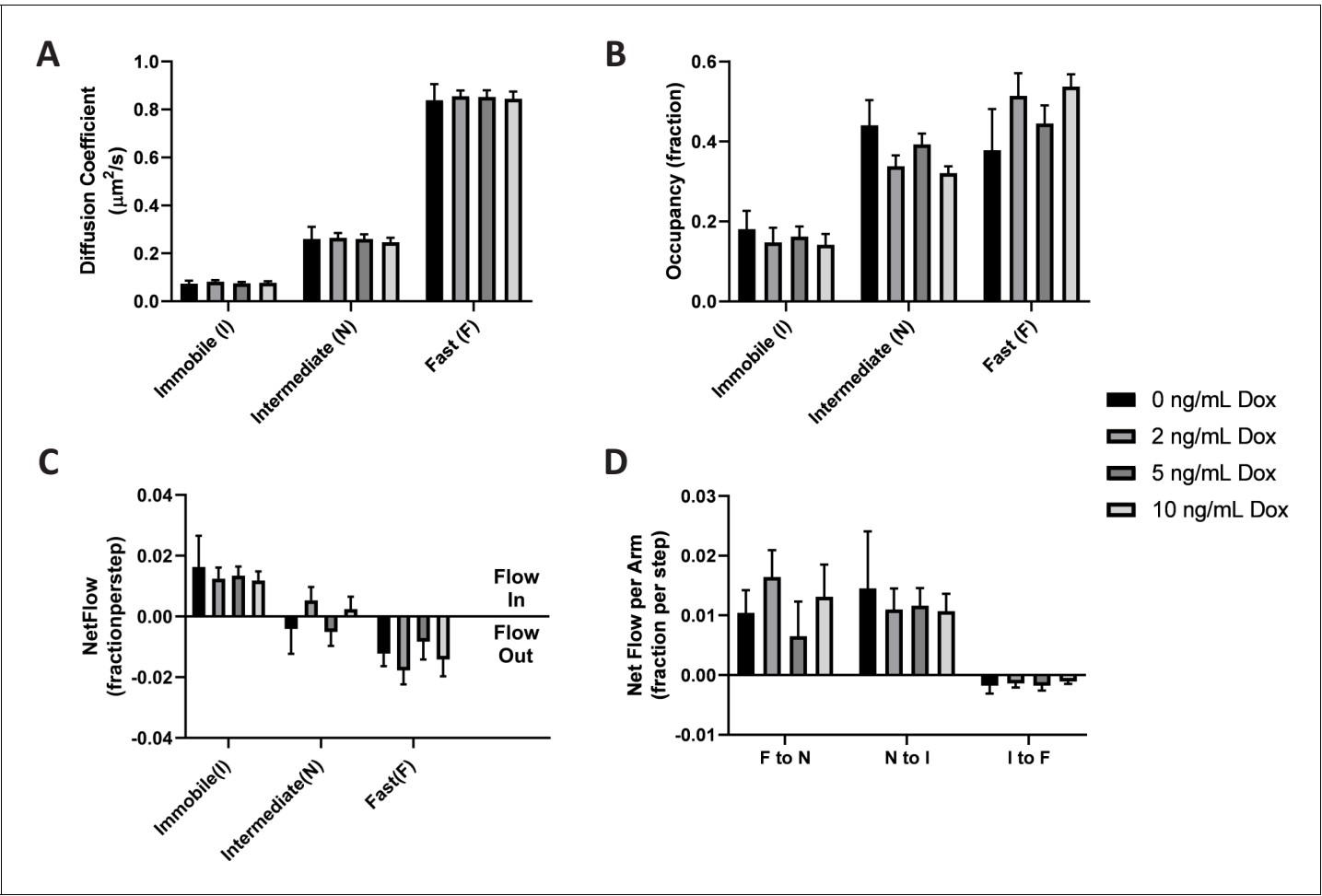

**Figure 6.** KRas[G12D] diffusion properties remain constant over a broad range of expression levels. Spt-PALM trajectories of KRas[G12D] were acquired at 12 ms/frame after inducing the cells at 0, 2, 5, and 10 ng/mL Dox for 36–48 hr, and the diffusion models were inferred as described previously using vbSPT. All aspects of the diffusion model discussed earlier, including diffusion coefficients (A), state occupancies (B), net mass flow per state (C), and net mass flow per arm (pair of states, (D) at the different Dox concentrations were analyzed and compared. Error bars represent 95% CIs (n = 12 for 0 ng/mL Dox, n = 22 for 2 ng/mL Dox, n = 30 for 5 ng/mL Dox, and n = 18 for 10 ng/mL Dox).

The online version of this article includes the following source data and figure supplement(s) for figure 6:

**Source data 1.** Excel sheet for data used for generating all the panels.
**Figure supplement 1.** vbSPT model outputs from KRas[G12D] diffusion trajectories acquired at different conditions (frame rate, total number of trajectories, and Dox concentration).
**Figure supplement 2.** A three-state model remains optimal for describing KRas[G12D] diffusion over a broad range of expression levels.
**Figure supplement 3.** Net mass flow between KRas[G12D] diffusion states is independent of expression level.

domains – is stable and independent of KRas$^{G12D}$ protein density on the membrane. This result coincides with the prior observation that the fraction of Ras in multimers remains constant at widely varying membrane densities (*Plowman et al., 2005*), implying that Ras multimer formation and nanodomain localization may be correlated processes.

## Discussion

Membrane nanodomains have been implicated in the regulation of many membrane-resident cellular processes such as Ras signaling (*Grecco et al., 2011*; *Staubach and Hanisch, 2011*; *Simons and Toomre, 2000*; *Schmick and Bastiaens, 2014*; *Varma and Mayor, 1998*; *Garcia-Parajo et al., 2014*), but studying the complex and heterogeneous membrane compartments in a living cell has remained a challenge. Using spt-PALM and detailed trajectory analysis, we were able to uncover rich details of how KRas$^{G12D}$ localizes and interacts with the membrane. Our results suggest that KRas$^{G12D}$ diffusion on the membrane is best recapitulated with a model that comprises three states – a fast state, an immobile state, and a previously unknown intermediate state. Leveraging the large number of diffusion trajectories, we were able to map the locations where KRas$^{G12D}$ exhibits specific diffusion states. These maps revealed membrane nanodomains corresponding to the intermediate and the immobile states of KRas$^{G12D}$. The intermediate nanodomains encompass the immobilization sites in a nested configuration, such that KRas$^{G12D}$ almost always transitions between the fast and the immobile states through the intermediate state. We also found that KRas$^{G12D}$ membrane diffusion is in a non-equilibrium steady state, with KRas$^{G12D}$ constitutively removed from the membrane through the immobile sites and replenished as fast diffusing molecules, potentially coupled to KRas$^{G12D}$ trafficking via endocytosis and recycling. Importantly, partitioning of KRas$^{G12D}$ into the three states remains invariant over a wide range of KRas$^{G12D}$ expression levels, demonstrating that KRas$^{G12D}$ diffusion and trafficking through the three mobility states and associated nanodomains is in a maintained, homeostatic condition. Together, these data start to paint a clear picture of the spatiotemporal dynamics of KRas$^{G12D}$ on the membrane, providing the basis for understanding the mechanisms of Ras multimer formation and signaling.

Based on these findings, we propose a new model for Ras membrane diffusion and trafficking as shown in *Figure 7*. In this model, Ras experiences at least three types of membrane environments: a 'regular' membrane region in which Ras freely diffuses with large step sizes, a 'transition zone' or intermediate domain with increased viscous drag and reduced step size, and within the latter an 'immobilization' site where Ras interacts with relatively static structures or molecules. Both the transition zones and the immobilization sites have finite lifetimes, some up to minutes, during which freely diffusing KRas$^{G12D}$ molecules could enter the transition zone, slow down, then either return to the fast state or become trapped at the immobilization sites. During entrapment, a fraction of the trapped KRas$^{G12D}$ molecules leaves the plasma membrane to enter the cycle of KRas$^{G12D}$ trafficking. This is in agreement with the current understanding that the removal of KRas from the membrane through endocytosis is concentration dependent and that the localization of KRas to the plasma membrane is an energy driven, PDEδ and Arl2-mediated enrichment process (*Schmick et al., 2014*; *Schmick et al., 2015*). Our work adds important details to this trafficking model in that the removal of KRas$^{G12D}$ from the plasma membrane likely occurs during the entrapment phase and its recycling primarily takes place in membrane regions conferring fast mobility. Additionally, the transient entrapment of KRas$^{G12D}$ could also provide an effective mechanism to locally concentrate Ras molecules to facilitate multimer formation, which arguably is a critical step for signaling (*Chen et al., 2016*; *Tian et al., 2007*; *Nussinov et al., 2019*). Thus, the various membrane nanodomains directly influence the mobility, trafficking, and potentially multimer formation and signaling of KRas$^{G12D}$, although details of the trafficking and multimer formation processes are yet to be defined.

The three-state diffusion model proposed in this study refines existing models of KRas$^{G12D}$ membrane diffusion by introducing a previously unresolved intermediate state and capturing the role of membrane nanodomains in KRas$^{G12D}$ diffusion. While heterogeneous diffusion properties of KRas$^{G12D}$ and other Ras isoforms have been reported, the prior studies lacked the throughput or spatiotemporal resolutions to determine whether two states, namely a fast diffusion state and an immobile state, are adequate to recapitulate KRas$^{G12D}$ diffusion on the membrane. With the diffusion model defined, we were able to subsequently demonstrate that the intermediate and immobile states of KRas$^{G12D}$ are each associated with a distinct membrane domain. The measured sizes of the

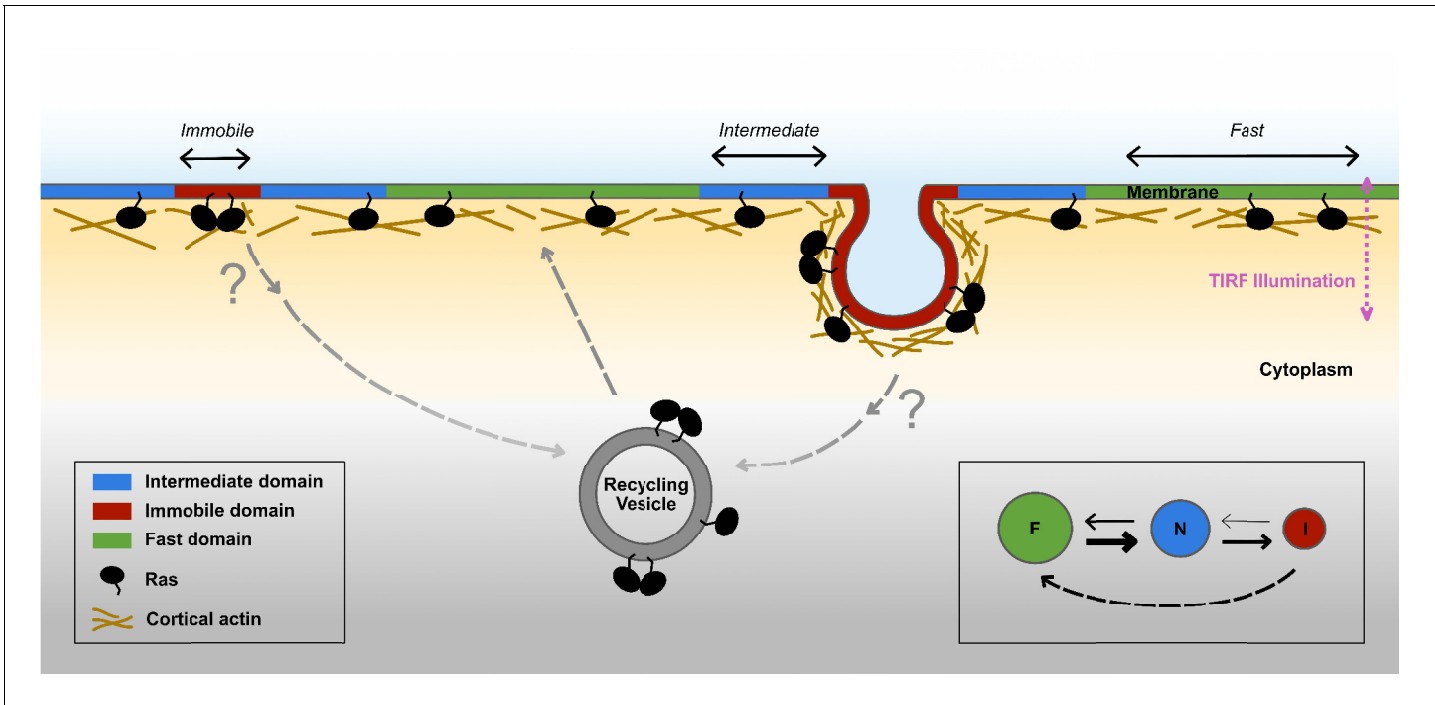

**Figure 7.** Proposed model for membrane nanodomains regulating KRas[G12D] mobility and trafficking. For KRas[G12D], the cell membrane comprises of at least three different compartments conferring each of the three diffusion states , namely the fast (and free), the intermediate, and the immobile diffusion states, depicted as green, blue, and red regions, respectively. The membrane compartments associated with the immobile and the intermediate states of KRas[G12D] are nanoscopic membrane structures. KRas[G12D] is continuously removed from the immobile state, some of which may be mediated via endocytosis. Internalized KRas[G12D] molecules are eventually transported back to the membrane as fast diffusing species through incompletely understood recycling processes. KRas[G12D] immobilization domains could locally enrich KRas[G12D] molecules to facilitate KRas[G12D] multimer formation and potentially signaling. The arrows in the inset (bottom right) indicate mass flows between each pair of states.

immobilization and the intermediate domains of KRas[G12D] were ~70 nm and ~200 nm, respectively, consistent with previous beliefs that nanoscopic membrane domains regulate Ras organization on the membrane. We note that, although a three-state model best fits our data, the model could still be an over-simplification. Among other possibilities, both endocytic and non-endocytic mechanisms may contribute to the immobilization of Ras but cannot be distinguished based on diffusion properties since Ras is immobile in both cases. In fact, there are also indications of more than one type of intermediate domains judging from the estimated domain size (*Figure 2F* and *Figure 2—figure supplement 1*).

An important feature of the model in *Figure 7* is that the membrane nanodomains associated with the immobile state of KRas[G12D] are surrounded by those associated with the intermediate state, creating a nested configuration between the two nanodomains. A plausible scenario is that the structures that trap KRas[G12D] preferentially form in the membrane regions enriched in certain proteins or lipids and/or more densely packed. In this scenario, KRas[G12D] would have to travel through the intermediate zone to access the immobilization structures, explaining the state transition pathway in *Figure 2A*. This scenario is also consistent with the observation that the intermediate domains are on average larger in size than the immobilization domains, and that the two nanodomains have similar lifetimes (to the extent of our temporal resolution). In support of this hypothetic scenario, a growing body of literature demonstrates the importance of phosphatidylserine in KRas clustering and activation (*Zhou et al., 2015*; *Cho et al., 2016*; *Zhou et al., 2017*; *Ariotti et al., 2014*). In addition to the KRas tail encoding for phosphatidylserine specificity, a significant fraction of phosphatidylserine display slow motion on the membrane as well (*Zhou et al., 2017*; *Kay et al., 2012*).

Aside from the steady state partitioning of KRas[G12D] in the different membrane domains, our data also offered important insight into the membrane dynamics of KRas[G12D]. We measured a constant flow of KRas[G12D] from the fast state to the immobile state. Without exchanging KRas[G12D] with

the cytosol, this directional flow would have caused net loss of KRas$^{G12D}$ from the fast state and accumulation in the immobile state as described in *Figure 4B–C*, yet the experimentally observed state configuration (*Figure 2A*) remained stable over time (*Figure 4A*). We therefore reasoned that KRas$^{G12D}$ needs to be constantly removed from the immobile state ('sink') and replenished via the fast state ('source'), potentially coupled to membrane trafficking such as endocytosis and recycling (*Schmick et al., 2014*; *Schmick et al., 2015*; *Roy et al., 2002*; *Rocks et al., 2005*; *Lu et al., 2009*), since previous studies have shown that endocytosis is a primary mechanism for KRas$^{G12D}$ removal from the plasma membrane (*Schmick et al., 2014*). In support of this, the lifetime of the immobilization domains was estimated to be on the order of 2–5 min on average (*Figure 3A–B*), which is typical of many endocytic systems (*Mayor et al., 2014*; *Kaksonen et al., 2005*). The exact mechanism of KRas$^{G12D}$ internalization, however, remains incompletely understood at present.

It is noteworthy that the spatial partitioning of KRas$^{G12D}$ and more generally the diffusion model were invariant over a broad range of KRas$^{G12D}$ expression levels, which coincides with previous observations where the clustered fraction of KRas or HRas was independent of the protein expression level (*Plowman et al., 2005*; *Tian et al., 2007*). This corroborates the idea that membrane partitioning of Ras and perhaps many other membrane resident molecules are in an actively maintained, homeostatic condition. This intriguing property of certain membrane proteins (*Plowman et al., 2005*; *Sharma et al., 2004*) has drawn much attention and led to at least two mechanistic models of multimer formation, one based on self-nucleation (*Plowman et al., 2005*) and the other driven by actomyosin (*Gowrishankar et al., 2012*). Both mechanisms assumed the different states of the protein on the plasma membrane to be in equilibrium. Our results argue that the mass exchange between the plasma membrane and the cytosol breaks the equilibrium and has to be taken into account in order to accurately model the partitioning behavior of membrane proteins. A clear, mechanistic understanding of this property is important to understand how Ras functions on the membrane, since the Ras multimers have been strongly implicated in signaling. Further experimental and computational work along this line is currently underway.

A fundamental albeit implicit result from the present study is the importance of experimental parameters in accurately determining the diffusion model, a critical step for in-depth analysis of protein dynamics on the membrane. While there are many different software packages for analyzing spt-PALM trajectories, the importance of controlling the particle density during image acquisition has not previously been recognized to our knowledge. Imaging at a per frame particle density of 0.05–0.1 per µm$^2$, which is typical for single-molecule localization microscopy, yielded varying estimated model parameters in our early attempts to track KRas$^{G12D}$ with spt-PALM (*Figure 1—figure supplements 1–5*). Using simulations, we found the source of variability to be a small fraction of misconnected trajectories mostly caused by fast moving molecules. In order to minimize the trajectory misconnection, we kept the density of activated PAmCherry1 in each frame to below 0.03 per µm$^2$ at an acquisition rate of 12 ms/frame (*Figure 1—figure supplements 2* and *3*). With this precaution, we were able to yield a highly consistent diffusion model from trajectories acquired in different cells and under different conditions. This was critical to defining a previously unresolved state with intermediate mobility (D ~ 0.3 µm$^2$/s) and to all subsequent analyses. We recommend the same precautions to be taken for studies of other membrane molecules.

In summary, our work sheds new light on how complex nanodomains organize on the membrane to dictate Ras diffusion and trafficking. The insights gained here offer useful guidance to future experiments that aim at determining the molecular and structural identities of the Ras-associated membrane nanodomains and defining the mechanisms of Ras multimer formation and signaling. The results demonstrate the utility of high-throughput SPT and trajectory analysis in uncovering rich details of the spatiotemporal dynamics of Ras on the membrane, which should be readily applicable to studies of other membrane molecules or processes in cellular compartments.

# Materials and methods

**Key resources table**

| Reagent type (species) or resource | Designation | Source or reference | Identifiers | Additional information |
|---|---|---|---|---|

*Continued on next page*

*Continued*

| Reagent type (species) or resource | Designation | Source or reference | Identifiers | Additional information |
|---|---|---|---|---|
| Cell line (*Homo-sapiens*) | U2OS | ATCC | RRID: CVCL_0042 ATCC Cat#: HTB_96 | Parent cell line for generating U2OS-tetR |
| Cell line (*Homo-sapiens*) | U2OS-tetR | This paper. | RRID: CVCL_XZ88 | Single U2OS clone stably expressing the tet repressor (tetR). |
| Cell line (*Homo-sapiens*) | U2OS-tetR PA-mCherry1-KRas G12D | This paper. | RRID: CVCL_XZ89 | Single U2OS cell clone stably expressing PAmCherry1-KRas$^{G12D}$ under Doxycycline regulation |
| Antibody | anti-RAS (mouse monoclonal) | Abcam | RRID: AB_941040 Abcam Cat#: ab55391 | Mouse monoclonal antibody |
| Antibody | anti-beta-Tubulin (mouse monoclonal) | Thermo Fisher | RRID: AB_86547 ThermoFisher Cat#: 32–2600 | Mouse monoclonal antibody |
| Software, algorithm | vbSPT | Perssonet al. | RRID: SCR_017554 | Algorithm for extracting diffusion parameters from SPT data |
| Software, algorithm | µManager | Invitrogen | RRID: SCR_000415 | Micro-manager open source microscopy platform |

## Cell culture

KRas$^{G12D}$ was genetically fused to PAmCherry1, a red fluorescent protein, to ensure high labeling specificity and efficiency. The PAmCherry1-KRas$^{G12D}$ coding sequence is placed under a CMV promoter regulated by the TetOn operon. The construct was transduced via lentivirus into an isogenic U2OS-tetR cell line (RRID:CVCL_XZ88) that constitutively expresses the tet repressor (tetR); the cell line was derived from the parent U2OS (human osteosarcoma, ATCC; RRID:CVCL_0042; verified via third party STR analysis). Single cell clones were subsequently isolated and screened to yield isogenic cell lines (RRID:CVCL_XZ89) that express the PAmCherry1-KRas$^{G12D}$ fusion protein under doxycycline (Dox) regulation. Cell lines were tested for mycoplasma regularly using standard in-house PCR test.

## Western blotting

Cells were cultured in 6-well plates for 24–48 hr before lysing with a RIPA buffer (Thermo Scientific, 89901) supplemented with an inhibitor cocktail (ThermoFisher, 88668). Cell lysates were then harvested, sonicated, and centrifuged. The supernatant is assayed using BCA and analyzed using a Bris-Tris gel (4–12%, ThermoFisher NP0323). Protein transfer was performed on a low fluorescence PVDF membrane (EMD Millipore, IPFL10100). The membrane was then immunostained for fluorescence detection using a Li-COR Odyssey. The antibodies used for this study were: KRas (mouse monoclonal, Abcam ab55391, RRID:AB_941040, used at 1:200 dilution), Tubulin (mouse monoclonal, Thermo-Fisher 32–2600, RRID:AB_86547, used at 1:500 dilution).

## Cell treatment for single particle tracking

Cells were grown in fluorobrite DMEM (Thermo Fisher Scientific A1896701) with 10% FBS in 8-well Lab-Tek chambers and Dox-induced for 1.5 days before imaging. Cells were serum starved for at least 12 hr prior to data acquisition.

Single-particle tracking was performed on a custom single-molecule localization microscopy setup, as previously described (*Creech et al., 2017*). Briefly, the setup was constructed around a Nikon Ti-U microscope, equipped with a high numerical aperture (NA) objective lens (Nikon 60x, NA = 1.49 oil immersion) for total internal reflection fluorescence (TIRF) imaging, lasers emitting at 405 nm (Coherent, OBIS) and 561 nm (Opto-Engine) for photoactivation and excitation, respectively, and an electron-multiplied charge-coupled display (EM-CCD, Andor iXon+) for single molecule

detection. All image acquisition was done using micro-manager (*Edelstein et al., 2014*) (RRID:SCR_000415) and processed using in-house Matlab scripts (*Creech et al., 2017*).

## Particle density optimization

We found that particle densities higher than 0.03 $\mu m^{-2}$ per frame under our experimental conditions (12 ms/frame with the fastest diffusion rate at ~1 $\mu m^2$/s) led to occasionally misconnected trajectories, and that even a small fraction of such misconnected trajectories could lead to incorrect model outputs with vbSPT (*Figure 1—figure supplements 2* and *3,* and *5*). In addition, the threshold for maximum displacement between adjacent frames also had an impact on trajectory misconnection, although to a lesser extent for the values tested using simulated trajectories (*Figure 1—figure supplements 2* and *3*). Thus, for diffusion model construction, we chose to use a high frame rate (12 ms/frame) and a low particle density (<0.03 $\mu m^{-2}$) to eliminate misconnected trajectory segments while maintaining a sufficient number of trajectories. However, it is beneficial to obtain more trajectories to accurately infer the model parameters with vbSPT (RRID: SCR_017554), especially for the transition probabilities (*Persson et al., 2013*). As demonstrated in *Figure 6—figure supplement 1*, the diffusion coefficients and the occupancies typically converged with only a few thousand trajectories, but the transition probabilities required significantly more trajectories to converge. Thus, we usually acquired spt-PALM data at higher particle densities once the model size has been defined; for these datasets, we could safely enforce a three-state model during vbSPT data analysis, since the diffusion model should not depend on the rates of frame acquisition rate and photoactivation. This strategy allowed more flexibility in spt-PALM data acquisition and robustness in the subsequent analyses.

## Trajectory connection for single particle tracking

We constructed single-molecule diffusion trajectories of PAmCherry1-KRas[G12D] by connecting the centroid positions of the same particles in successive frames. Particles in adjacent frames were deemed to be the same particle if their centroids were within a certain threshold distance. To define the threshold distance, we first constructed the trajectories using a large (~2,000 nm) distance, from which a step size histogram could be obtained (see *Figure 1—figure supplement 3*). The step size histogram from PAmCherry1-KRas[G12D] typically consists of two segments; signal and noise. The first segment comprises the signal with the first peak around ~70 nm and extending to ~500 nm, and all step sizes beyond ~500 nm was attributed to noise originating from misconnected trajectories generated by the unrealistically large threshold distance. Based on this histogram, we reconstructed the diffusion trajectories using 500 nm as the threshold distance for 12 ms frame acquisition, and 800 nm for 35 ms frame rate movies (using the same method). A new step size histogram was then obtained, which was essentially identical to the first segment of the original step size histogram, confirming that the new threshold distance eliminated most of the misconnected trajectories. The step size histograms of trajectories obtained under the same conditions were also highly consistent, allowing us to set the same threshold value for each condition. Trajectories were terminated if multiple particles were found within the threshold distance in the next frame. Further, all movies acquired at 12 ms frame rate had the additional constraint of having fewer than 0.03 particles/$\mu m^2$ for every frame to lower the chance of misconnecting two different particles in adjacent frames. Thus, all resulting trajectories were constructed without ambiguity.

## 2D Markov simulation

We relied on 2D simulations that mimic experimental observations for both experiment optimization and as controls for some of the analysis. Simulations were used to determine the thresholds used for trajectory synthesis (particle density threshold as shown in *Figure 1—figure supplement 2*, and connection distance threshold as shown in *Figure 1—figure supplement 3*), as well as a negative control to test the null hypothesis for spatial clustering (*Figure 2* and *3*) and equilibrium state analysis (*Figure 4*).

The inputs to the simulations were experimentally derived diffusion parameters: number of trajectories, diffusion coefficients, occupancies, transition matrix, frame rate, and the trajectory density. The trajectory density and the number of trajectories are used to determine the width of the simulation space. At the start of the simulation, every particle is randomly assigned a coordinate and a

state based on the occupancies. Once a state is assigned, particles are assigned new coordinates by drawing displacements for each dimension from the corresponding $X \sim N(0, 2Dt)$, where each state has a different diffusion coefficient. At the next time step, a new state is randomly assigned to every particle based on its current state and the transition probability matrix. This process is repeated for the total simulation time. When the simulation was used as the negative control (*Figures 2*, *3* and *4*), the simulation was run for every single movie acquired and the results were compared to the experiment.

## State assignment and averaging

States for each trajectory segment were assigned using vbSPT (contained in field est2.sMaxP, refer to the vbSPT manual). The state assignment is based on trajectory displacements, not the coordinates (e.g. if a trajectory has three coordinates, then two states are returned for the two steps). In order to prevent over counting for the pair correlation analysis (*Figure 2 and 3*), in the case of a single molecule staying in the same domain for multiple frames, we averaged all of the coordinates (including both ends) that were assigned the same state for consecutive time points in a single trajectory.

## Pair correlation function

Pair correlation function, or *g(r)*, in general, measures the deviation of the particle density from the expected value from a reference particle as a function of distance. More specifically, *g(r)* was calculated for each particle by counting the number of other particles within a circular shell at distance of r and r + 10 nm and dividing by the expected number of particles assuming uniform distribution. Therefore, when the observed number of particles for a given distance is equal to the expected number of particles given complete spatial randomness, *g(r)*=1 and signifies random distribution of particles. Accordingly, *g(r)*>1 indicates clustering behavior since there are more observed particles around each particle than expected, and *g(r)*<1 represents cases where there are fewer particles than expected. Every movie was sliced into non-overlapping time segments (1, 5, 10, 20 min) and the average position for each state segment was extracted (as described in State Classification and Averaging) such that every coordinate represented a continuous track for an individual particle in a domain. Therefore, the coordinates used to calculate the pair correlation function represented either different particles that visited the same domain or the same particle that left the domain and returned at a later time. The resulting coordinates were separated into each of the three states, and the *g(r)* was calculated for the coordinates of a given state within the given time slice. In cross pair correlation function analysis, *g(r)* was calculated for a given pair of different states.

## Statistical analysis

Sample size is shown for each figure in the figure captions as 'n' and was not predetermined. All results on model parameters and subsequent quantifications such as mass-flow rates are shown as arithmetic mean ±95% confidence interval. Spt-PALM datasets with insufficient number of trajectories to fully fit up to a 10-state model (e.g. *Figure 1E*) using vbSPT were discarded. The full raw dataset, including an outlier with abnormally long average trajectory length and all the discarded datasets are presented in *Figure 6—figure supplement 1*.

## Acknowledgements

The authors would like to thank many colleagues for their helpful discussions, including those at OHSU (Drs. Laura Heiser, Xubo Song, Molly Kulesz-Martin, Pamela Cassidy, and others) and at the Frederick National Laboratory for Cancer Research (Drs. Frank McCormick (also at UCSF), Thomas Turbyville, Dwight Nissley, and others). Research in the Nan lab was supported by startup funds from the Knight Cancer Institute at OHSU, the Damon Runyon Cancer Research Foundation, the M J Murdock Charitable Trust, and the Prospect Creek Foundation. XN is also supported by a Cancer Systems Biology Consortium (CSBC) grant (U54 CA209988, PI: Joe Gray) and by the Knight Caner Early Detection Advanced Research (CEDAR) Center at OHSU. YL is currently supported by OHSU Center for Spatial Systems Biomedicine (OCSSB). DMZ acknowledges support from the OCSSB and the National Science Foundation under grant MCB 1715823. We also thank Ms. Julia Shangguan

(Nan Lab) for graphic illustrations and Dr. Martin Lindén (University of Uppsala) for helpful discussions on vbSPT.

## Additional information

### Funding

| Funder | Grant reference number | Author |
| --- | --- | --- |
| National Institutes of Health | U54 CA209988 | Young Hwan Chang<br>Joe W Gray<br>Xiaolin Nan |
| National Science Foundation | MCB1715823 | Daniel M Zuckerman |
| Damon Runyon Cancer Research Foundation | | Xiaolin Nan |
| M.J. Murdock Charitable Trust | | Joe W Gray<br>Xiaolin Nan |
| Prospect Creek Foundation | | Joe W Gray<br>Xiaolin Nan |

The funders had no role in study design, data collection and interpretation, or the decision to submit the work for publication.

### Author contributions
Yerim Lee, Data curation, Software, Formal analysis, Validation, Investigation, Visualization, Methodology, Writing—original draft, Writing—review and editing; Carey Phelps, Data curation, Software, Formal analysis, Visualization, Methodology; Tao Huang, Data curation, Software, Formal analysis, Visualization; Barmak Mostofian, Formal analysis, Visualization, Methodology; Lei Wu, Kai Tao, Data curation, Investigation; Ying Zhang, Data curation; Young Hwan Chang, Software, Methodology; Philip JS Stork, Joe W Gray, Resources, Methodology; Daniel M Zuckerman, Supervision, Methodology, Writing—review and editing; Xiaolin Nan, Conceptualization, Resources, Supervision, Funding acquisition, Methodology, Writing—original draft, Project administration, Writing—review and editing

### Author ORCIDs
Yerim Lee (iD) https://orcid.org/0000-0002-0164-5551
Barmak Mostofian (iD) http://orcid.org/0000-0003-0568-9866
Joe W Gray (iD) http://orcid.org/0000-0001-9225-6756
Daniel M Zuckerman (iD) https://orcid.org/0000-0001-7662-2031
Xiaolin Nan (iD) https://orcid.org/0000-0002-0597-0255

### Decision letter and Author response
Decision letter https://doi.org/10.7554/eLife.46393.sa1
Author response https://doi.org/10.7554/eLife.46393.sa2

## Additional files

### Supplementary files
• Transparent reporting form

### Data availability
We have provided a complete set of model parameters derived from all raw single-particle tracking videos in the supplementary information.

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
