## [Decision Letter]

**Acceptance summary:**

Lee and colleagues utilize high throughput spt-PALM combined with rigorous analysis approaches to determine the diffusion characteristics of an active mutant of Kras on live cell membranes. They report three diffusion states and identified a previously unidentified intermediate mobile state. They provide evidence for distinct membrane domains corresponding to different diffusion states and show that the transition from the fast state to immobile state has to go through the intermediate state. They further suggest that KRas is depleted from the membrane via the immobile state and replenished to the fast state via endocytosis and recycling. This work provides several new insights into the membrane behaviors of KRas. The experimental approaches are thorough and rigorous.

**Decision letter after peer review:**

Thank you for submitting your article "High-throughput single-particle tracking reveals nested membrane nanodomains that dictate Ras diffusion and trafficking" for consideration by *eLife*. Your article has been reviewed by three peer reviewers, one of whom is a member of our Board of Reviewing Editors, and the evaluation has been overseen by Philip Cole as the Senior Editor. The reviewers have opted to remain anonymous.

The reviewers have discussed the reviews with one another and the Reviewing Editor has drafted this decision to help you prepare a revised submission.

Essential revisions:

1) KRas has two different isoforms (KRas4A and KRas4B) and various mutant forms. Considering that different KRas versions may have different diffusion characteristics, please include the specific information of the KRas construct used in this study and provide the rationale on focusing on the G12D mutant. Moreover, since the analysis is restricted to KRas^G12D^, this should be acknowledged in the title and Abstract.

2) The interpretation of the 3 states identified by the vbSPT analysis that underpin the model proposed in Figure 7 is unclear. Intuitively one would expect that a molecule that goes from a fast diffusion state to immobilization may transit through an intermediate diffusion state and the question arises to what extent the method and SPT imaging parameters influence the identification of the third intermediate state. The transition rates and the spatial correlation between the intermediate diffusion state and the slow/immobile state could also arise from analyzing trajectories without the necessary simultaneous high spatial and temporal resolution so that molecules transition from free to immobile appear to be in a third intermediate state due to the limited resolution. Can the authors exclude this possibility and ensure that the statement that the third state is 'an obligatory link' is not an overinterpretation?

3) What are the domains sizes when the data were taken with 12 ms/frame rates (subsection “KRas diffusion states correspond to distinct membrane domains”, third paragraph)? Are the data of the step angles (Figure 3D) consistent with domain sizes of <70 nm for immobile diffusion given the localization precision of the experiment?

4) The authors speculate that the transition between the intermediate into the immobile state could be caused by conformational changes or transient binding events (or both). However, both of these processes are much faster than the transition observed with spt-PALM. Further, there are alternative explanations for two spatially correlated diffusion states such as intermediate diffusion in membrane domains and protein-protein interactions within in these domains (immobile diffusion), interaction with cortical actin versus interaction with effector proteins etc.

5) Figure 5. The authors conclude KRas is in a non-equilibrium steady state (NESS). Could KRas at the plasma membrane be in an steady state equilibrium when one takes non-membrane bound KRAS into account that is not being detected? How is a NESS model consistent with the active mechanisms of multimer formation that is discussed later in the manuscript (subsection “KRas diffusion model is invariant over a range of expression levels”, first paragraph)?

6) The authors provide no evidence that the 'immobile domain' (i.e. the domains in which KRas is immobile, as distinct from the domains being immobile) are membrane invaginations as depicted in Figure 7.

7) To demonstrate that KRas is depleted from the membrane via the immobile state and replenished to fast state via endocytosis and recycling, experiments should be repeated with inhibition of the endocytic pathway.

8) The use of the dox-regulated expression system is good because this prevents over-expression problems. The data to support this is presented in Figure 1A. However, the endogenous and recombinant Ras proteins are presented in different strips – it is therefore not obvious how these should be compared. At a minimum, the figure legend text should state that the exposure times for the endogenous and recombinant Ras images were the same.

9) Are the transition rates influences by the (average) trajectory length i.e. the probability that a molecule undergoes transition? What is the average trajectory length?

---

## [Author Response]

Essential revisions:1) KRas has two different isoforms (KRas4A and KRas4B) and various mutant forms. Considering that different KRas versions may have different diffusion characteristics, please include the specific information of the KRas construct used in this study and provide the rationale on focusing on the G12D mutant. Moreover, since the analysis is restricted to KRas^G12D^, this should be acknowledged in the title and Abstract.

We completely agree and have updated the manuscript accordingly, including the title, the Abstract, the main text and all the figure legends, to reflect the specific isoform and mutant form used in the current work.

The reason that we focused on KRas^G12D^ in this study was two-fold. First, the G12D mutant is constitutively active, making it convenient to study potential connections between Ras diffusion and signaling via perturbations (e.g. dynamin inhibition) without having to stimulate the cells with serum or growth factors first. Second, from our unpublished preliminary results, KRas WT and G12D appear to be highly similar – if not identical – in both spatial organization and diffusion dynamics, thus results on KRas^G12D^ are actually representative of KRas properties in general. For considerations on the length and focus of this manuscript, we chose to not discuss KRas signaling or the comparison between KRas WT and G12D at this point; work in these directions is currently underway.

2) The interpretation of the 3 states identified by the vbSPT analysis that underpin the model proposed in Figure 7 is unclear. Intuitively one would expect that a molecule that goes from a fast diffusion state to immobilization may transit through an intermediate diffusion state and the question arises to what extent the method and SPT imaging parameters influence the identification of the third intermediate state. The transition rates and the spatial correlation between the intermediate diffusion state and the slow/immobile state could also arise from analyzing trajectories without the necessary simultaneous high spatial and temporal resolution so that molecules transition from free to immobile appear to be in a third intermediate state due to the limited resolution. Can the authors exclude this possibility and ensure that the statement that the third state is 'an obligatory link' is not an overinterpretation?

Thank you for bringing up this very important question, and our answer is yes – we are confident that the intermediate state is a true state and an obligatory link between the fast and the immobile states instead of an artefact of unresolved state mixing based on the evidence below,

- Spt-PALM data acquired at 35 ms/frame and those at 12 ms/frame gave rise to the same, 3-state diffusion model with consistent occupancy for all three states (Figure 2A and Figure 6—figure supplement 1). This suggests that the intermediate state is unlikely a result of insufficient time resolution that led to unresolved transitions between the fast and the immobile states, because we would have otherwise observed significantly higher occupancy for the intermediate state at 35 ms/frame. In fact, from the model presented in Figure 2A, it is clear that the transition probabilities between each pair of states were low (< 0.1 in each case), suggesting that state transitions were rare events compared with image acquisition, therefore the states could be clearly resolved at both 12 ms and 35 ms time resolutions.

- To also exclude the possibility that insufficient spatial resolution (or localization precision in this case) caused an artefactual ‘intermediate’ state, we have performed additional computer simulations to define the impact of localization error on vbSPT model output.

The details of the simulations could be found in the new Figure 1—figure supplement 4. Results from these simulations show that, even at localization errors as high as 100 nm, vbSPT will not yield a three-state model if the underlying diffusion follows two states (see Figure 1—figure supplement 4A). Of note, the single-molecule localization errors in our experiments were typically around 25 nm (at 35 ms/frame) to 40 nm (at 12 ms/frame). Thus, we do not think that the intermediate state was a result of insufficient spatial resolution (precision) either.

- We are also confident about the three-state model because of the consistency across the data acquired from different cells and even under different Dox concentrations, when spt-PALM data were acquired under optimal image acquisition conditions (Figure 1). As we also discussed in the manuscript, even when spt-PALM data were acquired under non-optimal conditions (e.g. particle density too high), at which point vbSPT model output would exhibit significant variability, the resulting diffusion coefficients for all the states still clustered around three values: ~0.05 µm^2^/s, ~0.25 µm^2^/s, and ~0.80 µm^2^/s. These results further suggest that KRas^G12D^ indeed exhibits three diffusion states.

- Last but not least, the three-state model was confirmed independently by Dr. Thomas Turbyville and colleagues at the NCI *RAS Initiative*, where they have used a different tagging strategy (the Halo-tag) that afforded single-molecule tracking at 5 ms/frame. This personal communication after we completed the manuscript and did not play a role in how we defined our diffusion model, it offers additional evidence for our conclusions.

In retrospect, we did not arrive at this consistent three-state model without substantial efforts. The major obstacle, as alluded above and also pointed out by the reviewer, was that the imaging and data processing parameters did have an impact on the consistency of the model output. That was why we focused on the importance of carefully controlling the data acquisition conditions in the first subsection in Results (with 6 figure supplements to Figure 1), with the hope to inform fellow scientists of the steps necessary to accurately define the diffusion model using spt-PALM and vbSPT.

As the reviewer also alluded to, the fact that the various datasets all led to a three-state model with define state transition path should not be just a coincidence. It may represent organizationally and biophysically plausible scenario for membrane proteins to transition from a fast (F) state to an immobile (I) state by going through an intermediate (N) state, which could otherwise be an abrupt change in mobility. While we do not have data for a large number of membrane proteins to certify the generality of this statement, our preliminary data indicate that the same applies to both KRAS WT. A recent study on b2-adrenergic receptor (Schwenzer et al., bioRxiv; http://dx.doi.org/10.1101/406488; posted on September 3, 2019) also revealed a similar scenario where the receptor transitions between the F state and the I state via an N state (see Figure 3 therein). Thus, the same three-state configuration may be more generally applicable to many other targets.

We have updated our manuscript based on the discussions above.

3) What are the domains sizes when the data were taken with 12 ms/frame rates (subsection “KRas diffusion states correspond to distinct membrane domains”, third paragraph)? Are the data of the step angles (Figure 3D) consistent with domain sizes of <70 nm for immobile diffusion given the localization precision of the experiment?

The average domain sizes for the intermediate and the immobile states were measured to be ~80 nm and ~100 nm when the images were acquired at 12 ms/frame (Figure 2F, inset), and hence our statement ‘… The distinction between the two domains became much *less* significant with data taken at 12 ms/frame. …’. Apologies for missing the word ‘*less*’ in our original text, which has now been corrected. Prompted by this question, we have also revised the following sentence ‘… which we attribute to the low localization precision of PAmCherry1 molecules at this fast frame rate …’ to ‘…which we attributed to the shorter trajectory durations (~50 ms at 12 ms/frame compared to ~175 ms at 35 ms/frame; see Figure 1—figure supplement 1), which in turn was due to the lower photon yield per frame from single PAmCherry1 molecules at this fast frame rate’.

The way that we estimated the domain size associated with a specific diffusion state of KRas^G12D^ was to measure the maximum distance between two localizations of KRas^G12D^ while in the state. Thus, the measured domain size depends on three factors: the actual domain size, the spatial precision, and lastly the efficiency of domain sampling. While moving within a membrane domain, PAmCherry1-KRas^G12D^ molecules report their locations to ‘paint’ the outline of the domain; the longer these molecules can be tracked, the better sampling of the domain is, and the more accurate the estimated domain size is. Thus, for domain size measurements, data taken at 35 ms/frame will be more reliable because of the better spatial precision and the longer duration of the trajectories (~175 ms on average at 35 ms/frame vs. ~50 ms on average at 12 ms/frame). In comparison, domain sizes measured from data taken at ~12 ms/frame will have smaller contributions from the actual motions of the molecules and instead be dominated by localization precision (~40 nm localization error, or 40*sqrt(2) ~ 56 nm error for point-to-point measurements) for both the immobilization and the intermediate domains. This resulted in a much smaller difference between the estimated sizes for the two domains from data taken at 12 ms/frame; it is also part of the reason why we combined information from both frame rates depending on the type of analysis.

The precision of step angle measurements (Figure 3D) will be affected by the localization error, particularly when the steps are short (relative to the localization error). That said, the localization error will add uncertainty to all measured step angles in an unbiased manner. Thus, the overall distribution of the step angle (not individual angles) would be similar to the true distribution, especially when the sample size is large as in our case (e.g. >50k for each domain with pooled data). This explains why we were able to observe clear differences in the step angle distributions for the fast, the intermediate, and the immobile states of KRAS^G12D^.

Based on this analysis, we think that the step angle distributions reflect the nature of the interactions between KRas^G12D^ and the respective membrane domains. In particular, the small size (<70 nm) of the immobilization domain is consistent with the observed enrichment of acute angles (Figure 3D). This is because the preference for acute angles is primarily associated with the domain boundary (if the molecules prefer to stay within the current domain instead of crossing over to another; it’d be the opposite if the molecules prefer to escape). By contrast, for steps inside the domain, there would be no preference for any particular angles. Thus, the smaller the domain, the higher probability the molecules will hit the boundary during diffusion, and the stronger the preference for acute angles (again, if the molecules prefer to stay within the current domain). This should also explain why the enrichment of acute angles for the intermediate domain is not as significant as the immobilization domain.

4) The authors speculate that the transition between the intermediate into the immobile state could be caused by conformational changes or transient binding events (or both). However, both of these processes are much faster than the transition observed with spt-PALM. Further, there are alternative explanations for two spatially correlated diffusion states such as intermediate diffusion in membrane domains and protein-protein interactions within in these domains (immobile diffusion), interaction with cortical actin versus interaction with effector proteins etc.

Apologies for the confusion. We did not mean to exclude other mechanisms that could give rise to the various diffusion states of KRAS^G12D^. Our work focused on the role of membrane organization – that is, the nanodomains – in governing the diffusion and trafficking of KRAS^G12D^. The mechanisms through which the nanodomains confer different mobilities for KRAS^G12D^ are currently unknown. As pointed out by the reviewers, conformational changes, transient binding (including that to proteins, lipids, or other types of molecules), and molecular packing, could all impact KRAS^G12D^ diffusion. Thus, KRAS^G12D^ diffusion state transitions can be viewed as domain crossing events potentially accompanied by conformational changes, transient binding events (including that with cortical actin), or other events.

We would also like to use this opportunity to clarify that, although conformational changes and transient binding events are expected to be much faster than our frame rate (12 ms/frame or slower), distinct states can be captured and resolved as long as the molecules stay in each state sufficiently long. For example KRAS^G12D^ stayed in each state for a few 100 ms on average (judging from the state transition rates in Figure 2A), thus the individual states can be captured at 12-35 ms/frame with little ambiguity. We continue to work on improving the spatial and temporal resolution of spt-PALM experiments, such that even more details during state transitions can be captured, to help address questions including whether there are multiple types of immobilization and/or intermediate domains.

5) Figure 5. The authors conclude KRas is in a non-equilibrium steady state (NESS). Could KRas at the plasma membrane be in an steady state equilibrium when one takes non-membrane bound KRAS into account that is not being detected?

Yes, we think that membrane-bound and cytoplasmic KRAS should be in an equilibrium steady state as a whole (unless there is significant loss of KRAS from this system via processes such as exocytosis).

While the sub-system of membrane-bound KRAS is in a non-equilibrium steady state, the whole system comprising both membrane-bound (detected by TIRF) and cytoplasmic (not detected by TIRF) KRAS should constitute an equilibrium steady state. Here, the cytoplasmic compartment serves to receive KRAS removed from the cell membrane (likely via the immobile state) and replenish KRAS to the cell membrane (at membrane locations that support fast diffusion), establishing a ‘cycle’ of KRAS trafficking through which an equilibrium can be maintained. In fact, the continuous exchange of Ras proteins between the plasma membrane and the cytoplasm has been observed on HRAS and NRAS (Rocks et al., 2005) and more recently on KRAS (Schmick et al., 2014; 2015). Our data and model are consistent with the previous reports on RAS trafficking.

How is a NESS model consistent with the active mechanisms of multimer formation that is discussed later in the manuscript (subsection “KRas diffusion model is invariant over a range of expression levels”, first paragraph)?

At present we do not have sufficient data to conclude on the relationship between the NESS model and the mechanisms of KRAS multimer formation. How RAS forms multimers still remains unclear and was only touched upon in our work. What our data revealed was that nanodomain localization and multimer formation are likely correlated processes, because the fraction of KRAS in multimers (from previous reports) and that in nanodomains (from our data) are both constant over a broad range of KRAS expression levels. In addition, the fact that membrane-bound KRAS^G12D^ is in a NESS begs for re-examination of previous models for RAS multimer formation, since previous active or passive models assumed an equilibrium steady state for membrane-bound RAS. This may change current views in the field that disfavor the passive mechanisms (which would have resulted in an expression level-dependent fraction of clustered RAS in an equilibrium system), but a more systematic investigation would be necessary to better define how RAS forms multimers.

6) The authors provide no evidence that the 'immobile domain' (i.e. the domains in which KRas is immobile, as distinct from the domains being immobile) are membrane invaginations as depicted in Figure 7.

There were indeed no data provided to prove that the KRas^G12D^ immobilization domains are membrane invaginations in Figure 7. For the scope of this work, we meant to depict some potential scenarios where KRAS could be internalized. In addition to the membrane invaginations, we also drew some immobilization domains as ‘flat’ membrane regions in the original Figure 7. These scenarios were included because previous literature reports suggested that although endocytosis is primarily responsible for KRAS internalization, non-endocytic mechanisms should not be ruled out (Schmick et al., 2014; 2015). The same papers described how recycling vesicles return KRAS to the plasma membrane. We did not mean to dive into this subject here other than citing literature.

To clarify the purpose of having Figure 7, we have modified the schematic to reflect the facts a) that both endocytic and non-endocytic compartments (both potentially linked to the KRAS immobilization domains) could be involved; and b) that the exact mechanisms through which KRAS^G12D^ is removed from the immobilization is currently unknown (hence the question marks). We have also softened the language in relevant sections.

7) To demonstrate that KRas is depleted from the membrane via the immobile state and replenished to fast state via endocytosis and recycling, experiments should be repeated with inhibition of the endocytic pathway.

Analyzing KRAS^G12D^ diffusion in the presence of endocytosis inhibitors is a great suggestion. We actually performed such experiments but chose not to include the data in this manuscript because we would like to focus on the nested nanodomains and the NESS model. For the importance of this issue, we would like to discuss some of our preliminary results (Author response image 1). These results also suggest that identifying the sites of KRAS immobilization, internalization, and recycling would require a larger body of work (which is currently underway). For the current work, we have modified the text to clearly state ‘KRAS^G12D^ immobilization sites are linked to endocytosis’ as a hypothesis based on literature evidence instead of a conclusion at present.

**Author response image 1. respfig1:** Effect of dynamin inhibition (with Dyngo 4a) on KRas^G12D^ diffusion. Shown in the plots were diffusion constants for all detected states by vbSPT. Untreated cells (left) were used as control (similar to that shown in Figure 1—figure supplement 5C). Right, U2OS-PAmCherry1-KRas^G12D^ cells were treated with Dyngo 4a (50 µM) for 30 min prior to spt-PALM and imaged for another 30 min before switching to a new sample.

In preliminary studies, we found that treatment with a dynamin inhibition (dyngo 4a) trapped all KRAS^G12D^ in the immobile state (see Author response image 1). After dynamin treatment, KRAS^G12D^ accumulates on the membrane (data not shown) and interestingly, the majority of KRAS^G12D^ molecules become immobile. Dynamin is involved in many types of endocytic vesicles. Thus, the trapping of KRAS^G12D^ in the immobile state could suggest that a) stopping KRAS^G12D^ internalization cuts off recycling but not accumulation of KRAS to the immobile state; and b) that pre-endocytic membrane structures may be sufficient to trap KRAS^G12D^.

Using correlative superresolution microscopy, we also obtained preliminary evidence that KRAS^G12D^ may interact with multiple types of potentially ‘static’ membrane structures, some of which membrane vesicles adherent to the cortical actin, potentially serving as KRAS immobilization sites (https://doi.org/10.1017/S1431927619006834). We expect this complete body of work will be shared with the community in the near future.

8) The use of the dox-regulated expression system is good because this prevents over-expression problems. The data to support this is presented in Figure 1A. However, the endogenous and recombinant Ras proteins are presented in different strips – it is therefore not obvious how these should be compared. At a minimum, the figure legend text should state that the exposure times for the endogenous and recombinant Ras images were the same.

The strips in the original western blot (as in the original Figure 1A) did have the same exposure, which led to the estimation that exogenous KRAS^G12D^ and endogenous KRAS levels were comparable at 2 ng/mL Dox. Prompted by this question, we have re-examined the western blot results and decided not to make this statement based on existing results.

Specifically, in reviewing the original data, we realized that there was a technical glitch that made all the bands appear as doublets (including the RAS bands), which should not have been the case based on our past experience and literature. We thus have re-run the western blot to correct for the issue. For the new western blot, we had to use a different RAS antibody (Abcam ab55391) because the original antibody had recently been discontinued (sc-521; product link https://www.scbt.com/p/k-ras-2b-antibody-c-19?requestFrom=search). The relative abundance between the endogenous KRAS and PAmCherry1-KRAS^G12D^ appeared to be different from what we observed using sc-521. The origin of this difference is unclear to us at present, but it could be due to differences in specificity (Waters et al., Sci. Signal., v10, 2017; http://doi.org/10.1126/scisignal.aao3332). In light of this, we have changed the text to reflect the fact the western blots were only semi-quantitative in quantifying the relative abundance of the endogenous vs. exogenous KRAS, particularly when the isoform specificity of the antibodies is not as well defined.

9) Are the transition rates influences by the (average) trajectory length i.e. the probability that a molecule undergoes transition? What is the average trajectory length?

When using vbSPT to analyze spt-PALM data, the average trajectory length does have an impact on the extracted model parameters. In general, the longer the average trajectory, the more accurate the model parameters are (Persson et al., 2013). Thus, it is better to use datasets with longer average trajectory length. That said, the effect seems modest and only becomes significant at trajectory lengths of >100 (data points), which is impractical in most spt-PALM experiments using fluorescent proteins.

For example, for PAmCherry1-KRAS^G12D^ spt-PALM, the average trajectory lengths were ~4 frames and ~5 frames for data acquired at 12 ms/frame and 35 ms/frame, respectively (Figure 1—figure supplement 1) under our imaging conditions. These values were typical of spt-PALM experiments using fluorescent proteins (see the Persson et al. reference above). The vbSPT algorithm was actually developed to specifically address this challenge (among others); it is able to resolve events at a time scale as long as 10x the average trajectory length, thanks to the stochastic nature of the events, the large number of trajectories, and the presence of a reasonable number of long trajectories. Therefore, longer SPT trajectories could indeed reveal additional events not captured by our current experimental configuration.

To investigate how the average trajectory length would impact the transition rates, we have taken a group of datasets (12 ms/frame; minimum 10,000 trajectories) and deleted all trajectories shorter than 5 frames. Of these, 6 datasets remained to still contain a sufficient number of trajectories (4,000 – 7,500 trajectories each) for vbSPT to run successfully. These correspond to about ~1/4 of the original number of trajectories in each case. Analyzing the filtered datasets yielded a similar diffusion model (see Author response image 2), demonstrating a net mass flow within the system in the same direction and with similar amplitudes. Also similar are the lack of transitions between the F state and the I state as well as the roughly balanced in-and-out flows of the N state.

**Author response image 2. respfig2:** Diffusion model (left) and net mass flow analysis (right) using filtered spt-PALM data after removing trajectories shorter than 5 frames.

However, we need to point out that, while 4,000-7,500 trajectories are typically sufficient for vbSPT to converge on the model size and the diffusion constants, they are usually NOT sufficient for vbSPT to converge on the transition rates. Thus, despite the similarities between the resulting model (Author response image 2) and that shown in Figure 2A, there are some differences in the transition rates. With the limited information at hand, it is difficult to assess which approach is better for ensuring accurate transition rate computation – using long trajectories, or a larger number but somewhat shorter trajectories. Since it is usually difficult to increase the trajectory length, the common approach is to try use a larger number of shorter trajectories as we did in this work. That said, we are experimenting with other tagging strategies (e.g. SNAP and Halo tags) so organic fluorophores can be used to allow for acquisition of longer trajectories.